# Over-parameterized Student Model via Tensor Decomposition Boosted Knowledge Distillation

**Yu-Liang Zhan**
Gaoling School of Artificial Intelligence
Renmin University of China
`zhanyuliang@ruc.edu.cn`

**Zhong-Yi Lu**
School of Physics
Renmin University of China
`zlu@ruc.edu.cn`

**Hao Sun**[*]
Gaoling School of Artificial Intelligence
Renmin University of China
`haosun@ruc.edu.cn`

**Ze-Feng Gao**[*]
School of Physics
Renmin University of China
`zfgao@ruc.edu.cn`

## Abstract

Increased training parameters have enabled large pre-trained models to excel in various downstream tasks. Nevertheless, the extensive computational requirements associated with these models hinder their widespread adoption within the community. We focus on Knowledge Distillation (KD), where a compact student model is trained to mimic a larger teacher model, facilitating the transfer of knowledge of large models. In contrast to much of the previous work, we scale up the parameters of the student model during training, to benefit from over-parameterization without increasing the inference latency. In particular, we propose a tensor decomposition strategy that effectively over-parameterizes the relatively small student model through an efficient and nearly lossless decomposition of its parameter matrices into higher-dimensional tensors. To ensure efficiency, we further introduce a tensor constraint loss to align the high-dimensional tensors between the student and teacher models. Comprehensive experiments validate the significant performance enhancement by our approach in various KD tasks, covering computer vision and natural language processing areas. Our code is available at `https://github.com/intell-sci-comput/OPDF`.

## 1 Introduction

Large-scale pre-trained models are gradually achieving remarkable milestones due to the exhibit of remarkable performance across various tasks [1–7]. These models leverage extensive pre-training data and parameters, enabling them to effectively encapsulate a significant breadth of world knowledge [8, 9] and exhibit strong generalization capabilities across diverse tasks [1, 10–13]. Following this trajectory, the utilization of increased data and parameters has emerged as a notable trend in enhancing the performance of pre-trained models in recent years, leading to the number expansion of pre-trained model parameters from millions to billions [4, 14, 15].

Despite their impressive performance, the substantial storage demands and high computational complexity hinder the practical deployment of these models in real-world applications. Therefore, on the one hand, some studies focus on pre-training relatively smaller models (such as BERT-base-uncased [2]) on domain-specific or task-specific corpora [16–18]. However, due to the lesser over-parameterization of small models compared to large ones, their generalization capability often

---

[*]Corresponding authors.

38th Conference on Neural Information Processing Systems (NeurIPS 2024).

falls short, resulting in suboptimal fine-tuning performance on downstream tasks. On the other hand, model compression methods, such as pruning less informative parameters [19–21] or utilizing *knowledge distillation* (KD) [22] to transfer knowledge from larger models (teachers) to smaller ones (students), have been proposed. KD has swiftly diversified into numerous branches, primarily falling into two categories: *i.e.,* logits-based [22–26] and features-based [27–30] depending on the source of student model knowledge. Nevertheless, as student models have fewer trainable parameters and limited capacity, a significant performance gap remains between student and teacher models.

To address the disparity between small and large models, this study aims to over-parameterize small student models as large ones during distillation training to enhance their generalization capability. Typically, most parameters of student models are stored as matrices. Through tensor decomposition techniques [31–34] (*e.g.,* Singular Value Decomposition), each matrix can be factorized into a set of matrices, effectively increasing the total number of parameters during distillation. Moreover, after convergence, the factorized matrices can be merged to reorganize the parameter matrix of the student model. This paradigm leverages the benefits of over-parameterization during training without increasing the inference latency of student models.

However, incorporating tensor decomposition into over-parameterizing student models poses two major concerns that must be addressed. First, the potential information loss caused by tensor decomposition should be minimized, as small computation errors may accumulate and propagate exponentially within the stacked layers of student models. Second, in the over-parameterized student models, there is no effective mechanism to ensure the consistency of information between student and teacher models. Therefore, it is essential to choose appropriate tensor decomposition methods and design loss functions for high-order tensors to ensure the effective transfer of information from teacher to student models.

To address the above issues, we introduce the matrix product operator (MPO) [34] technique as the tensor decomposition strategy. The MPO decomposition, widely used in quantum many-body physics, efficiently factorizes any matrix with arbitrary dimensions into a set of higher-dimensional tensors, which can reconstruct the original matrix in almost lossless conditions [34–37]. These advantages make MPO an ideal method for over-parameterizing student models during distillation. Based on MPO, we also devise high-order tensor alignment losses for student and teacher models to ensure the effective transfer of information in tensor representation.

Therefore, in this paper, we propose a general Over-Parameterization Distillation Framework, namely **OPDF**, to improve the performance of knowledge distillation. Given the parameter matrices of a student model, we first over-parameterize them through MPO decomposition and then utilize high-order tensor alignment losses to ensure efficient information transfer. This framework only modifies the distillation training process, making it applicable to various student models and natural language processing (NLP) and computer vision (CV) tasks. We conduct extensive experiments in both NLP and CV domains. Experimental results demonstrate that our OPDF significantly enhances the effectiveness of model distillation, *e.g.,* improving BERT-base KD +1.6 on average. Moreover, our approach also enables the student model to achieve performance nearly on par with the teacher model, *e.g.,* AD-KD+Ours (83.4) *v.s.* BERT-base (83.4) in average metric on GLUE.

## 2 Related work

**Large Scale Pre-trained Models**    Large-scale pre-trained models have achieved remarkable success in many fields (*e.g.,* natural language processing (NLP) [38] and computer vision (CV) [15, 39]). Since the introduction of the Transformer architecture [40], the pre-training and fine-tuning paradigm in NLP, exemplified by models like BERT [2] and T5 [4], has shown outstanding performance across multiple tasks. Furthermore, the emergence of models like GPT-3 has demonstrated that increasing model size can significantly improve performance on low-resource tasks [12]. In the field of computer vision, models based on Transformers, such as ViT [7], have also performed exceptionally well. In our research, we improve the distillation process by increasing the parameters during the training phase of the student model, without introducing additional inference latency to the student model.

**Knowledge Distillation**    Knowledge Distillation (KD) methods are commonly used to compress models by transferring knowledge from a larger *teacher model* to a smaller *student model*. Building upon the initiative work by [22], the researchers have exploited the logits follows up with different

techniques in the computer vision field, *e.g.,* minimizing KL-Divergence (DKD [25]) or a Pearson correlation (DIST [26]). Logit-based methods have been also popular in NLP [41, 42]. Features-based methods have tried to align the features from intermediate layers of teacher and student models and minimize the differences [43]. After the intermediate representations have been introduced [27], a mount of features-based KD methods have been proposed to match the features, such as LGTM [44], DBKD [45] and AD-KD [46]. However, the capacity gap between the teacher and student models makes it difficult to imitate the hidden representations of the teacher [47]. Different from these existing KD methods, our proposed OPDF has utilized MPO decomposition to over-parameterize the student model in the training procedure to improve the student model generalization capability, which can minimize the capacity gap efficiently.

**Matrix Product Operators** Matrix Product Operators (MPOs) [34, 48], also known as tensor-train operators (TTOs) [33], have been proposed for a more efficient representation of the linear structure of neural networks [49, 50]. A large number of typical applications have utilized MPO-based methods to compress linear layers [51] and convolutional kernels [52] in the parameter matrices of deep models. Furthermore, existing works have applied the MPO method for lightweight fine-tuning of ALBERT [35], the efficient expansion for the MoE framework [36], the over-parameterization tuning process for PLMs [37], construct efficient PLM architecture [53, 54] and compressing datasets [55]. Unlike existing methods, our approach focuses on utilizing MPO decomposition to map parameters from low-dimensional spaces to high-dimensional spaces, to over-parameterize the student model during the distillation process, allowing the student model to benefit from more parameters and achieve better distillation results.

## 3 Preliminary

**Tensor Product** We denote a tensor $\mathcal{T}_{i_1, i_2, \ldots, i_n}$ as an array with $n$ indices, where $\{i_1, i_2, \ldots, i_n\}$ denotes the dimensions of the $n$ indices, respectively. In this manner, a vector (*i.e.,* $\boldsymbol{v}$) can be considered a 1-order tensor, while a matrix (*i.e.,* $\mathbf{W}$) can be regarded as a 2-order tensor. Consider $\psi_1, \ldots, \psi_p$ and $\phi_1, \ldots, \phi_q$ as the orthonormal bases of tensors $\mathcal{T}^{(1)}$ and $\mathcal{T}^{(2)}$, respectively. The tensor product, denoted as $\otimes$, can be obtained through the contraction of $\mathcal{T}^{(1)}$ and $\mathcal{T}^{(2)}$. Formally, the tensor contraction of $\mathcal{T}^{(1)} = \sum_{i=1}^{p} a_i \psi_{i_1}$ and $\mathcal{T}^{(2)} = \sum_{j=1}^{q} b_j \phi_{i_2}$ is defined as follow:

$$\mathcal{T}^{(1)} \otimes \mathcal{T}^{(2)} = \sum_{i=1}^{p} \sum_{j=1}^{q} a_i b_j \psi_{i_1} \otimes \phi_{i_2}. \tag{1}$$

The set $\psi_{i_1} \otimes \phi_{i_2}$ constitutes the orthonormal basis of the resulting vector Hilbert space, with the dimensionality of this Hilbert space being the product (i.e., $p \times q$) of $\mathcal{T}^{(1)}$ and $\mathcal{T}^{(2)}$.

**Tensor Decomposition** Tensor decomposition can be viewed as the reverse operation of the tensor product. A commonly employed approach is the singular value decomposition (SVD) algorithm. Given a tensor $\mathcal{T} \in \mathbb{R}^{i_1 \times \cdots \times i_n}$, the SVD operation performed $n$ times can decompose this tensor into $n$ local tensors $\mathcal{T}^{(k)}{}_{k=1}^{n}$. Conversely, the decomposed tensors can reconstruct the original tensor by sequentially applying the tensor product operator.

## 4 Method

In this section, we describe our proposed over-parameterized distillation framework. We first outline our approach, then introduce the details of matrix product operator decomposition and the over-parameterized student model strategy, and finally present the tensor alignment loss.

### 4.1 Overview

Current distillation methods primarily enhance the performance of student models by introducing constraints on logits or features between the student and teacher models. In contrast to these methods, our approach not only utilizes tensor decomposition to over-parameterize the student model for performance improvement but also designs alignment loss functions for the decomposed high-order tensors to further enhance the performance of the student model. To achieve this goal, we employ a

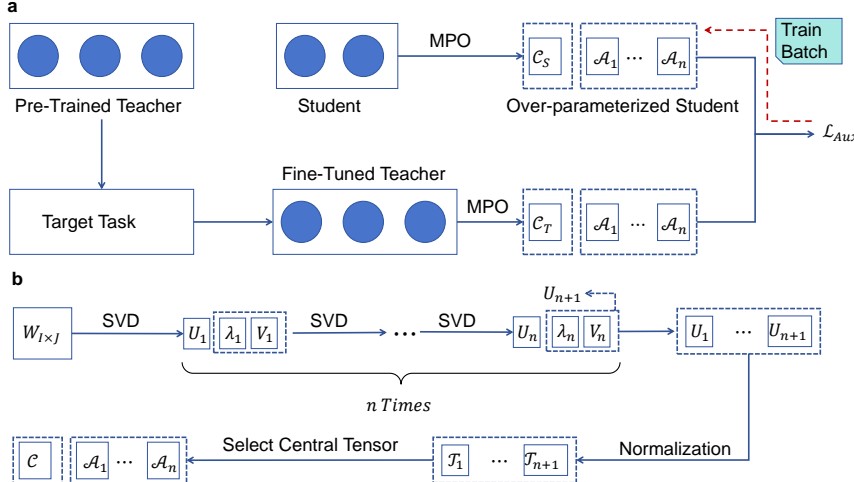

Figure 1: The overview of over-parameter distillation framework (OPDF) for knowledge ditillation. **a**, We use MPO decomposition to realize the over-parameter procedure for the student model. The auxiliary tensors of the student model are trained to imitate the auxiliary tensors of the teacher model closely. **b**, We present an illustrative example of MPO decomposition. A parameter matrix $\mathbf{W}_{I \times J}$ is decomposed into central tensor and auxiliary tensors.

tensor decomposition method to decompose the parameter matrices of the teacher and student models into a series of high-order tensor products. These high-order tensors can be used to reconstruct the original parameter matrices while significantly increasing the number of trainable parameters in the student model. After reconstruction, the student model has the same number of parameters as the original matrix without increasing inference time and model size. Additionally, by introducing distillation loss functions to allow the student model to learn from the teacher model in tensor representation, the effectiveness of knowledge distillation is further enhanced.

In our proposed over-parameterized distillation framework, we integrate a tensor decomposition strategy based on MPO into the student model to enlarge the parameter matrix (Section 4.2). Furthermore, we design a tensor alignment loss function to enhance the performance of the student model in the context of knowledge distillation (Section 4.3). An overview of our approach is depicted in Figure 1. We also provide a detailed description of our over-parameterized distillation framework in Algorithm S.1.

## 4.2 Over-paramterization Distillation Framework via MPO Decomposition

To leverage the advantages of over-parameterization during knowledge distillation, our method utilizes the MPO, a tensor decomposition technique that increases the number of model parameters. In this part, we initially present the specifics of the MPO method and subsequently outline its adaptation for over-parameterizing the student model.

**Matrix Product Operator Decomposition** The MPO decomposition is an efficient algorithm capable of factorizing a parameter matrix $\mathbf{W} \in \mathcal{R}^{I \times J}$ into a sequential product of multiple tensors [34]. Formally, given a matrix $\mathbf{M} \in \mathbb{R}^{I \times J}$, its MPO decomposition into a product of $n$ local tensors can be represented as:

$$\text{MPO}\,(\mathbf{M}) = \prod_{k=1}^{n} \mathcal{T}_{(k)}[d_{k-1}, i_k, j_k, d_k]. \tag{2}$$

The tensor $\mathcal{T}(k)[d_{k-1}, i_k, j_k, d_k]$ is a 4th-order tensor with dimensions $d_{k-1} \times i_k \times j_k \times d_k$, where $\prod_{k=1}^{n} i_k = I$, $\prod_{k=1}^{n} j_k = J$, and $d_0 = d_n = 1$. To link two sequence tensors, we have adopted the concept of a *bond* following the work of [48]. The bond dimension $d_k$ is defined by:

$$d_k = \min \left( \prod_{m=1}^{k} i_m \times j_m, \ \prod_{m=k+1}^{n} i_m \times j_m \right). \tag{3}$$

From Eq. (3), we can see that $d_k$ will be large in the middle and small on both sides. Following [35], we refer to the tensor right in the middle as *central tensor*, and the rest as *auxiliary tensor*. Figure 1(b) presents the illustration of MPO decomposition. You can find additional descriptions of tensors and MPO in Appendix A.

**Over-parameterzing Student Model.** Utilizing the MPO method within the framework of knowledge distillation, our objective is to extend the parameter scale of the student model, capitalizing on over-parameterization. More specifically, we can employ the MPO method to break down a portion of the parameter matrices into multiple tensors as illustrated in Eq. (2). Following MPO decomposition, the parameter count of the matrix $\mathbf{W}$ will increase based on the values of $\{d_k\}_{k=1}^{m}$, $\{i_k\}_{k=1}^{m}$, and $\{j_k\}_{k=1}^{m}$. The precise augmentation in parameter count, denoted as $N_{add}$, can be computed as follows:

$$N_{add} = \sum_{k=1}^{m} i_k j_k d_{k-1} d_k - \prod_{k=1}^{m} i_k j_k. \tag{4}$$

Therefore, during the knowledge distillation procedure, we can adopt MPO on student model parameter matrices to generate their corresponding multiple tensors. In this way, we can scale up the total parameter of the number of the student model without increasing its inference time consumption. After training the over-parameterized student model to convergence, we will perform tensor contraction on these decomposed tensors, to reconstruct the parameter matrices of the student model in almost lossless conditions which is detailed in Appendix B. This new student model has the same parameter number and inference latency as the original one and has benefited from over-parameterization during training.

### 4.3 Assisted Constraints for Knowledge Distillation

**Revisiting Prediction Match of Knowledge Distillation** Traditional knowledge distillation involves two stages: fine-tuning the teacher model for a specific task, followed by training strategies to constrain the student model to closely approximate the teacher model. These processes aim to transfer the knowledge from the teacher to the student model. Recent studies have mainly focused on directly learning from the features and logits of the teacher model to transfer crucial knowledge [23, 56].

However, these methods are limited by the capacity of the student model due to the limitation of total parameters. Moreover, this distillation approach based on cross-entropy loss constraints may lead to the student model *losing its ability to learn independently*. We aim to design a novel model distillation framework to enable the student model not only to effectively learn the knowledge from the teacher model but also to maintain its ability to learn independently.

**Distillation Loss for Auxilary Tensors.** To achieve the goal of "learning knowledge from the teacher model while maintaining the ability of the student model to learn independently," we introduce a high-order tensor alignment training method based on the MPO decomposition. A crucial merit of MPO decomposition is its ability to reorganize and aggregate the core information, decomposing the weight matrices into a central tensor (containing a large number of parameters and important information) and auxiliary tensors (containing fewer parameters and additional information to the central tensor) [35, 36]. Therefore, in the knowledge distillation, in addition to minimizing the cross-entropy loss concerning the ground truth, we add a loss constraint for aligning the auxiliary tensors between the student and teacher models:

$$\mathcal{L}_{Aux} = \frac{1}{n} \sum_{k=1}^{n} \text{MSE} \left( \mathcal{A}_{s,k}, \mathcal{A}_{t,k} \right), \tag{5}$$

where the matrices $\mathcal{A}_{t,k}$ and $\mathcal{A}_{s,k}$ refer to the auxiliary tensor of student and teacher models with the same dimensions respectively. MSE means the mean-square error loss function. To ensure that the student model learns from the teacher while preserving its central tensor for independent learning, we minimize the mean-square error loss between the auxiliary tensors of both the student and teacher models. Since this distillation framework is based on improvements to the weight matrices, it is

orthogonal to most current distillation methods. Therefore, it can further enhance the distillation effectiveness based on existing distillation methods (as thoroughly discussed in the experimental section). Hence, it can be widely applied to various knowledge distillation models.

## 5  Experiments

In this section, we assess the efficacy of our approach within two renowned domains: computer vision and natural language processing. Notably, the OPDF is designed to complement existing distillation techniques. Consequently, we apply our proposed OPDF with various standard distillation methods to validate its effectiveness. In the subsequent section, we detail our experimental setup's datasets and baseline methods. We then present the primary results achieved with the OPDF and provide a thorough analysis. Furthermore, we examine the influence of the degree of over-parameterization, MPO strategy and the learning rate on the performance of OPDF. We report the memory and time cost of experiments in Appendix D.1.

### 5.1  Experimental Setup

**Datasets and Metrics**  For NLP tasks, we evaluate our approach on text classification tasks in GLUE benchmark [57]. The tasks encompassed in our evaluation include RTE, MRPC, STS-B, CoLA, SST-2, QNLI, QQP, and MNLI. To facilitate comparison with baselines, we employ the F1 score and accuracy as metrics for MRPC and QQP, Matthew's correlation coefficient for CoLA, and the average of Pearson and Spearman correlations for STS-B. Accuracy is used as the metric for the remaining tasks, with the result for MNLI reported as the average across the matched (MNLI-m) and mismatched (MNLI-mm) domains. Additionally, we calculate the average score across all tasks to provide a comprehensive performance measure. In the context of CV tasks, we have applied the OPDF to the distillation of Vision Transformers (ViT) for image classification [7]. This was done using the ImageNet-21k dataset [58], ImageNet-1k, ImageNet Real [59], and ImageNet V2 [60] datasets. For these datasets, we use accuracy as the primary evaluation metric.

**Baseline Methods**  For NLP tasks, we implement OPDF on previous KD methods: BERT-of-Theseus [56], LGTM [44], DBKD [45] and AD-KD [46]. We replicated the baselines using the publicly released code to assess their performance on the test set. Additionally, LGTM was not previously evaluated across all tasks in its original publications, and we have addressed this omission using the provided code. It is important to note that DBKD is designed to estimate logits from decision distributions [45], and therefore we do not report performance on the STS-B task. For all experiments in natural language processing, we demonstrate the effectiveness of our method during the fine-tuning stage. We implement the teacher model as the fine-tuned "BERT-base-uncased" model [2]. In the context of CV tasks, TinyViT [61], which introduces a rapid pre-training framework, has emerged as a classical distillation method for ViT. The original paper on TinyViT discusses three versions of the model with varying parameter counts: TinyViT-5M, TinyViT-11M, and TinyViT-21M. To incorporate high-order tensor alignment loss into the distillation phase, we utilize CLIP-VIT-L/14 [7, 62], a variant of ViT, as the teacher model in our experiments. To assess the efficacy of OPDF, we pretrain the distillation model on ImageNet-21k and evaluate its linear probe performance on ImageNet-1k, ImageNet Real, and ImageNet V2, without any fine-tuning. During the pre-training stage, we adhere to the same experimental settings as described in the original paper. Furthermore, we juxtapose our method with SVD [32], a traditional tensor decomposition technique viable for over-parameterizing student models. Concretely, we employ SVD to substitute MPO within our framework and execute over-parameterization across all parameter matrices of the student model during knowledge distillation. Appendix D.2 shows more experimental details.

### 5.2  Main Experimental Results

**NLP Tasks**  We present the results on BERT in Table 1. Firstly, it is evident that integrating KD with over-parameterization methods yields the most significant performance enhancements. Over-parameterization enhances the generalization ability of the student model. Upon comparing the two tensor decomposition techniques, we find that MPO consistently outperforms SVD. This discrepancy arises from the singular value-based SVD in a two-dimensional space, limiting its ability to substantially increase model parameters compared to MPO decomposition (*e.g.,* 90M

Table 1: Comparison of performance on the GLUE benchmark (in percent). The terms "+SVD" and "+OPDF" represent the use of different over-parameterization methods in a KD model. "# Train Params" and "# Inference Params" refer to the total number of parameters during training and inference, respectively. Numbers marked with * indicate tasks not tested in the original studies; results here are reproduced from the published code. The best result for each task is highlight in bold. For all the results, we report the mean values of five runs using different random seeds.

| Datasets | RTE Acc. | MRPC F1/Acc. | STS-B Corr. | CoLA Mcc. | SST-2 F1/Acc. | QNLI Acc. | QQP F1/Acc. | MNLI Acc. | Avg. | # Train Params (M) | # Inference Params (M) |
|---|---|---|---|---|---|---|---|---|---|---|---|
| BERT-base [2] | 70.5 | 86.5/81.8 | 86.6 | 54.2 | 92.0 | 91.2 | 88.0/91.0 | 84.2 | 83.4 | 110 | 110 |
| **BERT-of-Theseus [56]** | | | | | | | | | | | |
| None | 65.5 | 85.3/79.6 | 86.2 | 39.2* | 90.4 | 88.7 | 86.1/89.6 | **81.5** | 79.2 | 66 | 66 |
| +SVD | 65.5 | 85.4/80.0 | 86.5 | 43.1 | 90.6 | 88.6 | 86.2/89.7 | 80.3 | 79.6 | 90 | 66 |
| +OPDF (Ours) | **66.2** | **85.9/80.5** | **88.6** | **45.2** | **91.3** | **89.0** | **86.8/90.2** | 81.4 | **80.5** | 160 | 66 |
| **LGTM [44]** | | | | | | | | | | | |
| None | 63.3 | 86.3/80.1 | 82.9* | 33.9* | 91.1 | **89.3** | **88.0/91.1** | **82.2** | 78.8 | 67 | 67 |
| +SVD | 64.7 | 86.8/81.9 | 83.1 | 37.4 | 91.2 | 88.6 | 86.5/89.4 | 79.3 | 78.9 | 91 | 67 |
| +OPDF (Ours) | **66.9** | **87.8/82.4** | **83.3** | **38.9** | **91.5** | 88.7 | 87.0/90.2 | 80.9 | **79.8** | 163 | 67 |
| **DBKD [45]** | | | | | | | | | | | |
| None | 61.2 | 83.3/75.5 | / | 25.2 | 88.1 | 86.1 | 85.3/88.7 | 76.1 | 74.4 | 53 | 53 |
| +SVD | 64.7 | 86.5/78.6 | / | 26.4 | 88.8 | 85.8 | 85.5/89.0 | 76.5 | 75.8 | 69 | 53 |
| +OPDF (Ours) | **69.1** | **88.4/83.3** | / | **27.2** | **89.8** | **86.5** | **86.9/90.2** | **77.7** | **77.6** | 83 | 53 |
| **AD-KD [46]** | | | | | | | | | | | |
| None | 68.8 | 88.7/84.3 | **89.3** | 53.1 | **91.5** | 90.8 | 85.9/89.5 | 81.7 | 82.4 | 67 | 67 |
| +SVD | 69.4 | 89.3/85.8 | 88.8 | 53.5 | 89.9 | 90.1 | 86.4/89.8 | 81.5 | 82.6 | 91 | 67 |
| +OPDF (Ours) | **71.7** | **90.3/86.8** | 88.9 | **55.0** | 91.3 | **91.1** | **86.8/90.0** | 82.1 | **83.4** | 182 | 67 |

*vs.* 160M in BERT-of-Theseus). In contrast, MPO allows for arbitrary scaling by increasing the order of decomposition, rendering it more suitable for over-parameterization. Secondly, following the integration of the OPDF method, the performance of prior KD techniques (BERT-of-Theseus, LGTM, DBKD, and AD-KD) have exhibited enhancements across a majority of tasks (*e.g.,* RTE, MRPC, CoLA, QQP), while maintaining comparability with the original method in other tasks. This highlights the versatility of OPDF, demonstrating its effectiveness across diverse models and a wide range of tasks. Finally, our findings have revealed that employing the OPDF method can even outperform the performance of the teacher model in MRPC and RTE datasets. This indicates that the process of over-parameterization endows the student model with stronger generalization capabilities, suggesting that employing over-parameterization may offer a potential solution to the bottleneck in current distillation methods where the performance of the student model fails to surpass that of the teacher model.

**CV Tasks** All CV results of our proposed method are shown in Table 2. We apply OPDF on three kinds of TinyVit with different total parameters. It is clear that with OPDF, the performance of TinyVit can be significantly improved. In particular, in all datasets, TinyVit applied OPDF is better than vanilla TinyVit. Moreover, TinyVit utilized OPDF with 11M parameters can achieve better performance than TinyVit with 21M parameters. It demonstrates that OPDF is an orthogonal method for various KD methods based on the Transformer whether in the CV or NLP field. Note that since we only involved the over-parameterization procedure in the training phase, the total parameter of the student model will not change in the inference phase. This merit makes the OPDF unique from the existing KD method: one would not increase inference time while enhancing model accuracy and enabling the model to acquire more knowledge from the teacher model. Moreover, we can observe that the performance of the original TinyVit, SVD over-parameterization, and OPDF over-parameterization improves as the number of parameters gradually increases. This indicates that compared to SVD, the MPO decomposition, which can decompose the parameter matrix to any size, can better enhance the expressive capacity of the student model. The impact of the over-parameterization scale on distillation effectiveness will be analyzed in detail in Section 5.3.

## 5.3 Further Analysis

**Performance Comparison *w.r.t.* Parameter Increasing Rate.** Our OPDF method facilitates the flexible expansion of model parameters, thereby highlighting the significance of the parameter increase rate on model performance. Consequently, we investigate the influence of this rate on model

Table 2: The linear probe performance (in percentage) of TinyViT, pre-trained on ImageNet-21k, ImageNet-1k [58], ImageNet Real [59], and ImageNet v2 [60]. Numbers marked with * indicate that these results are got by official checkpoint and released code. For all the results, we report the mean values of five runs using different random seeds.

| Datasets | Imagenet-1k | | Imagenet Real | | Imagenet V2 | | # Train Params | # Inference Params |
|---|---|---|---|---|---|---|---|---|
| | top-1 | top-5 | top-1 | top-5 | top-1 | top-5 | (M) | (M) |
| CLIP-ViT-L/14 [62] | 84.8* | / | 88.9* | / | 75.1* | / | 321 | 321 |
| **TinyVit-5M [61]** | | | | | | | | |
| None | 77.4* | 94.1* | 86.1* | 97.5* | 66.8* | 87.6* | 5.4 | 5.4 |
| +SVD | 77.9 | 95.1 | 86.3 | 97.3 | 68.7 | 88.4 | 7.6 | 5.4 |
| +OPDF (Ours) | **80.0** | **96.7** | **87.4** | **98.1** | **69.4** | **88.9** | 9.9 | 5.4 |
| **TinyVit-11M** | | | | | | | | |
| None | 80.5* | 95.6* | 87.8* | 98.0* | 70.7* | 90.4* | 11 | 11 |
| +SVD | 82.0 | 96.7 | 88.4 | 97.9 | 71.7 | 91.4 | 17 | 11 |
| +OPDF (Ours) | **82.5** | **96.9** | **88.9** | **98.3** | **72.4** | **92.6** | 23 | 11 |
| **TinyVit-21M** | | | | | | | | |
| None | 82.3* | 96.3* | 88.9* | 98.3* | 73.0* | 91.9* | 21 | 21 |
| +SVD | 82.9 | 96.8 | 88.3 | 97.8 | 71.8 | 92.4 | 29 | 21 |
| +OPDF (Ours) | **84.0** | **97.5** | **89.4** | **98.4** | **74.9** | **93.4** | 38 | 21 |

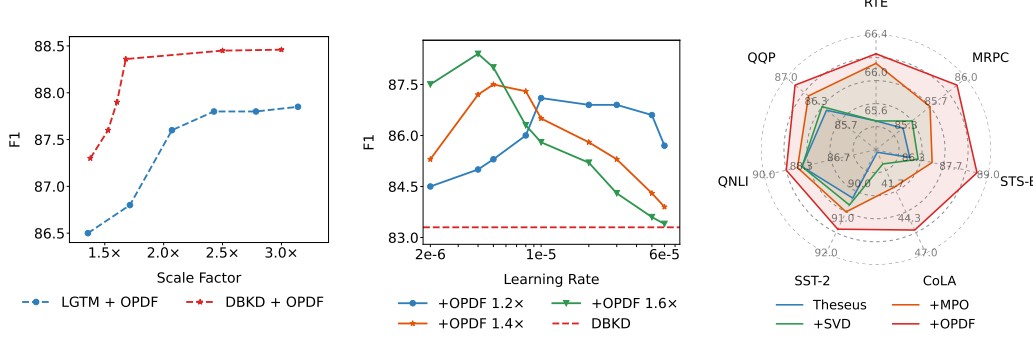

(a) Impact of scale factor    (b) Impact of learning rate    (c) Impact of OPDF components

Figure 2: The impact of over-parameterization scale, learning rate, and various components of the OPDF on distillation model performance is explored. Figure 2(a) demonstrates the performance of the LGTM and DBKD model on the MRPC task following the implementation of the OPDF. Figure 2(b) presents the performance of DBKD + OPDF with different over-parameterization scales on the MRPC task. Figure 2(c) displays the performance of the theseus model across various tasks, utilizing different over-parameterization methods and integrating various components of the OPDF.

efficiency further. To underscore the general applicability of our findings, we intentionally over-parameterize two models: DBKD and LGTM. We then elucidate their relationship with fine-tuning performance on MRPC tasks. All results are depicted in Figure 2(a).

It is observed that the performance of both the LGTM [44] and DBKD [45] models on the MRPC task consistently improves with an increase in parameters. This enhancement substantiates the efficacy of using the OPDF for over-parameterizing models, which in turn significantly boosts the performance of knowledge distillation models. Furthermore, after over-parameterization, the performance of the models is capable of achieving, at a minimum, the level of their original benchmarks (*e.g.,* 83.3 for DBKD and 86.3 for LGTM). The enhancement of model performance through over-parameterization has its limitations. As demonstrated in Figure 2(a), beyond certain thresholds of over-parameterization (*e.g.,* 1.6× for DBKD and 2.5× for LGTM), the performance improvements of the models no longer exhibit significant gains. This observation indicates that there are inherent limits to the benefits that can be achieved through over-parameterization in knowledge distillation models. These limits are likely influenced by structural characteristics of each model and size of the initial model configuration.

**Hyper-parameters Tuning** OPDF decomposes the original weight tensor through over-parameterization, leading to the updating of more parameters. Consequently, the tensor product results in larger updates to the existing parameters in the backward phase. In Figure 2(b), we illustrate

the relationship between the performance on the MRPC task and learning rate when the parameters of the DBKD model are expanded to $1.2\times$, $1.4\times$, and $1.6\times$ their original size.

There exists an optimal learning rate for every scale of over-parameterization. Deviating from this optimal rate, whether by increasing or decreasing the learning rate, results in diminished model performance. The reduction in performance due to a lower learning rate can be attributed to the model becoming trapped in a local optimum.

Additionally, we observe that peak model performance consistently increases with the scale of over-parameterization. This finding aligns with the conclusions drawn from Figure 2(a). Moreover, as the scale of over-parameterization increases, the learning rate required to achieve optimal model performance decreases. This occurs because using the tensor product to restore the shape of the tensors to that of the original weight tensors also scales the updated values, resulting in significant changes. Consequently, an increasing learning rate leads to declining performance in the KD model, indicating that the learning rate should decrease as the over-parameterization scale increases.

Finally, despite changes in the learning rate, the performance of the model with OPDF consistently remains at least as high as that of the original method. This indicates that OPDF is not sensitive to learning rate variations during the distillation stage. This resilience is due to OPDF's ability to factorize the parameter matrix in almost lossless conditions, ensuring that the decomposed matrix can match or exceed the training effectiveness of the original matrix without introducing errors.

**Impact of MPO strategy**  To demonstrate the robustness of our MPO methods, we applied different MPO methods to the DBKD and AD-KD model on the RTE, MRPC, STS-B, CoLA, and SST-2 task. The experimental results are presented in Table 3. To maintain consistent over-parameterization scales, we used the same decomposition scale (L) for each KD model across the same task.

Table 3: Comparison of performance on the GLUE benchmark (in percent). In the tensor representation, "L" denotes the number of "1"s in the dimension list.

| Experiments | RTE Acc. | MRPC F1/Acc. | STS-B Corr. | CoLA Mcc. | SST-2 F1/Acc. |
|---|---|---|---|---|---|
| **DBKD** | | | | | |
| $L$ | 4 | 8 | / | 7 | 4 |
| $\mathcal{T}^{32,L,24}_{64,L,48}$ | 69.1 | 88.4/83.3 | / | 27.2 | 89.8 |
| $\mathcal{T}^{16,2,L,2,12}_{32,2,L,2,24}$ | 68.0 | 86.3/81.0 | / | 25.2 | 89.0 |
| $\mathcal{T}^{4,4,2,L,2,3,4}_{8,4,2,L,2,4,6}$ | 68.5 | 87.9/82.5 | / | 26.1 | 88.9 |
| **AD-KD** | | | | | |
| $L$ | 8 | 3 | 6 | 1 | 3 |
| $\mathcal{T}^{32,L,24}_{64,L,48}$ | 71.7 | 90.3/86.8 | 88.9 | 55.0 | 91.3 |
| $\mathcal{T}^{16,2,L,2,12}_{32,2,L,2,24}$ | 70.9 | 89.6/86.1 | 88.7 | 54.4 | 89.2 |
| $\mathcal{T}^{4,4,2,L,2,3,4}_{8,4,2,L,2,4,6}$ | 71.0 | 89.8/86.4 | 88.3 | 54.9 | 90.4 |

We can observe that the performance of our approach consistently stabilizes around certain values, indicating that our method is not sensitive to the specific MPO techniques used. Therefore, when over-parameterizing, we should focus primarily on the decomposition scale rather than the MPO method employed.

**Ablation Study**  Our approach consists of two novel improvements: (1) the over-parameterization procedure for the student model, (2) the distillation loss for auxiliary tensors for effective training. To verify the effectiveness of each component, we conduct the ablation study on the GLUE benchmark to analyze the contribution of each part. We consider removing over-parameterization and distillation loss respectively. The ablation results of our OPDF are shown in Figure 2(c).

Firstly, it is clear that regardless of the over-parameterization method used, the area of the radar chart is greater than that of the vanilla theseus. This outcome suggests that over-parameterization can greatly improve the performance of distillation methods. Secondly, further analysis of the different over-parameterization methods reveals that MPO consistently outperforms SVD across all datasets. This improvement is attributed to MPO's ability to decompose parameter matrices into higher orders, effectively enlarging the size of the parameter matrix. Lastly, we examine the contribution of the $L_{Aux}$ term. The radar chart area is significantly larger when OPDF is utilized in conjunction with $L_{Aux}$ than with MPO alone. This indicates that $L_{Aux}$ effectively enhances knowledge transfer from

the teacher model. The underlying reason for this phenomenon is that over-parameterized models can concentrate on learning central tensors containing critical information, while the $L_{Aux}$ term assists in aligning auxiliary tensors. We can see that removing any component would lead to a decrease in the model performance. It shows the effectiveness of all these components in our approach.

# 6    Conclusion

In this paper, we proposed OPDF, a novel over-parameterization distillation framework designed to enhance the effectiveness of knowledge distillation. This framework employs MPO as a tensor decomposition technique to expand small models into larger ones, thereby bridging the capacity gap between the teacher and student models. Moreover, to enhance the effectiveness of knowledge distillation, our proposed OPDF framework introduces a tensor constraint loss. The OPDF framework utilizes MPO to decompose each weight matrix into a central tensor and auxiliary tensors. By aligning the auxiliary tensors, OPDF not only facilitates the transfer of crucial knowledge from the teacher model but also preserves the student model's ability to think independently. This approach provides the student model with the potential to outperform the teacher model. Our ablation studies demonstrated that all components of the OPDF contribute to enhancing the effectiveness of knowledge distillation. Experimental results across various tasks in natural language processing and computer vision domains validate the efficacy of our proposed method in improving model distillation. Although the number of parameters was increased by MPO during training, the factorized matrices can be merged to reorganize the original parameter matrix in almost lossless conditions. This means that OPDF can enhance the performance of the distillation model without increasing the inference latency. Moreover, since OPDF is based on tensor decomposition, it is orthogonal to most distillation methods.

In our future work, we will investigate more efficient and effective tensor decomposition methods for student model over-parameterization. In addition, we will also apply OPDF to other important backbone models, such as in the multimodal learning domains.

## Impact statement

This paper proposes a novel knowledge distillation framework for model compression field, which is helpful to reduce storage requirements and computational complexity. This method facilitates the practical deployment of models in real-world applications and supports energy conservation. We focus exclusively on over-parameterizing small student models, presenting no potential ethical risks.

## Acknowledgement

This work was supported by the National Natural Science Foundation of China (No. 62476278, No. 62206299, No. 92270118, and No. 11934020) and the Beijing Natural Science Foundation (No. 1232009). H.S would like to acknowledge the support from the Fundamental Research Funds for the Central Universities (No. 202230265).

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

# APPENDIX

## A  Tensor and Matrix Product Operators

As introduced in [34], the concept of a tensor is specified as:

**Definition1**
(Tensor). Let $D_1, D_2..., D_N \in N$ denote index upper bounds. A tensor $\mathcal{T} \in \mathbb{R}^{D_1,...,D_n}$ of order $N$ is an $N$-way array where elements $\mathcal{T}_{d_1,d_2,...,d_n}$ are indexed by $d_n \in \{1, 2, ..., D_n\}$ for $1 \leq n \leq N$

**Definition2**
(Matrix product operator). We can reshape a matrix to high order tensor, denoted as:
$$\mathbf{M}_{x \times y} = \mathbf{M}_{i_1 i_2 ... i_n, j_1 j_2 ... j_n} \tag{S.1}$$
Here, the one-dimensional coordinate $x$ of the input signal $\mathbf{x}$ with dimension $N_x$ is reshaped into a coordinate in a $n$-dimensional space, labeled by $(i_1 i_2 \cdots i_n)$. Hence, there is a one-to-one mapping between $x$ and $(i_1 i_2 \cdots i_n)$. Similarly, the one-dimensional coordinate $y$ of the output signal $\mathbf{y}$ with dimension $N_y$ is also reshaped into a coordinate in a $n$-dimensional space, and there is a one-to-one correspondence between $y$ and $(j_1 j_2 \cdots j_n)$. If $I_k$ and $J_k$ are the dimensions of $i_k$ and $j_k$, respectively, then
$$\prod_{k=1}^{n} I_k = N_x, \quad \prod_{k=1}^{n} J_k = N_y. \tag{S.2}$$

The MPO representation of $M$ is obtained by factorizing it into a product of $n$ local tensors
$$M_{i_1 \cdots i_n, j_1 \cdots j_n} = \mathcal{T}^{(1)}[i_1, j_1] \cdots \mathcal{T}^{(n)}[i_n, j_n] \tag{S.3}$$
where $\mathcal{T}^{(k)}[j_k, i_k]$ is a $D_{k-1} \times D_k$ matrix with $D_k$ the virtual basis dimension on the bond linking $\mathcal{T}^{(k)}$ and $\mathcal{T}^{(k+1)}$ with $D_0 = D_n = 1$.

## B  Theorem

**Theorem 1.** *Suppose that the tensor $\mathbf{W}^{(k)}$ of matrix $W$ that is satisfy*
$$\mathbf{W} = \mathbf{W}^{(k)} + \mathbf{E}^{(k)}, D(\mathbf{W}^{(k)}) = d_k,$$
$$where \quad ||\mathbf{E}^{(k)}||_F^2 = \epsilon_k^2, k = 1, ..., d - 1. \tag{S.4}$$
*Then $MPO\,(\mathbf{W})$ with the $k$-th bond dimension $d_k$ upper bound of truncation error satisfy:*
$$||\mathbf{W} - MPO\,(\mathbf{W})||_F \leq \sqrt{\sum_{k=1}^{d-1} \epsilon_k^2} \tag{S.5}$$

$Proof.$ The proof is by induction. For $n = 2$ the statement follows from the properties of the SVD. Consider an arbitrary $n > 2$. Then the first unfolding $\mathbf{W}^{(1)}$ is decomposed as
$$\mathbf{W}^{(1)} = \mathbf{U}_1 \lambda_1 \mathbf{V}_1 + \mathbf{E}^{(1)} = \mathbf{U}_1 \mathbf{B}^{(1)} + \mathbf{E}^{(1)} \tag{S.6}$$
where $\mathbf{U}_1$ is of size $r_1 \times i_1 \times j_1$ and $||\mathbf{E}^{(1)}||_F^2 = \epsilon_1^2$. The matrix $\mathbf{B}_1$ is naturally associated with a $(n-1)$-dimensional tensor $\mathcal{B}^{(1)}$ with elements $\mathcal{B}^{(1)}(\alpha, i_2, j_2, ..., i_n, j_n)$, which will be decomposed further. This means that $\mathbf{B}_1$ will be approximated by some other matrix $\hat{\mathbf{B}}_1$. From the properties of the SVD it follows that $\mathbf{U}_1^T \mathbf{E}^{(1)} = 0$, and thus

$$||\mathbf{W} - \mathcal{B}^{(1)}||_F^2$$
$$= ||\mathbf{W}_1 - \mathbf{U}_1 \hat{\mathbf{B}}_1||_F^2$$
$$= ||\mathbf{W}_1 - \mathbf{U}_1(\hat{\mathbf{B}}_1 + \mathbf{B}_1 - \mathbf{B}_1)||_F^2$$
$$= ||\mathbf{W}_1 - \mathbf{U}_1 \mathbf{B}_1||_F^2 + ||\mathbf{U}_1(\hat{\mathbf{B}}_1 - \mathbf{B}_1)||_F^2 \tag{S.7}$$

and since $\mathbf{U}_1$ has orthonormal columns,

$$||\mathbf{W} - \mathcal{B}^{(1)}||_F^2 \leq \epsilon_1^2 + ||\mathbf{B}_1 - \hat{\mathbf{B}}_1||_F^2. \tag{S.8}$$

and thus it is not difficult to see from the orthonormality of columns of $\mathbf{U}_1$ that the distance of the $k$-th unfolding ($k = 2, ..., d_k - 1$) of the $(d-1)$-dimensional tensor $\mathcal{B}^{(1)}$ to the $d_k$-th rank matrix cannot be larger than $\epsilon_k$. Proceeding by induction, we have

$$||\mathbf{B}_1 - \hat{\mathbf{B}}_1||_F^2 \leq \sum_{k=2}^{d-1} \epsilon_k^2, \tag{S.9}$$

combine with Eq. (S.8), this complets the proof.

## C    Algorithms

The over-parameterized distillation framework algorithm is shown in Algorithm S.1.

---

**Algorithm S.1** Over-parameterized distillation framework.

---

**Input:** The parameter matrices list of student model $\{\mathbf{M_s}(\mathbf{k})\}_{\mathbf{k=1}}^{\mathbf{n}}$, the parameter matrices list of teacher model $\{\mathbf{M_t}(\mathbf{k})\}_{\mathbf{k=1}}^{\mathbf{m}}$.

1:  **for** $k = 1 \rightarrow n$ **do**
2:      Select $\mathbf{M_t}(\mathbf{k_t})$ which has same shape as $\mathbf{M_s}(\mathbf{k})$.
3:      $\mathbf{M_s}(\mathbf{k}) \rightarrow \text{MPO}\,(\mathbf{M_s}(\mathbf{k}))$.
4:      $\mathbf{M_t}(\mathbf{k_t}) \rightarrow \text{MPO}\,(\mathbf{M_t}(\mathbf{k_t}))$.
5:  **end for**
6:  **repeat**
7:      Compute $L_{Aux}$ between $\{\text{MPO}\,(\mathbf{M_s}(\mathbf{k}))\}_{\mathbf{k=1}}^{\mathbf{n}}$ and $\{\text{MPO}\,(\mathbf{M_t}(\mathbf{k_t}))\}_{\mathbf{k_t=1}}^{\mathbf{m_t}}$ by using Eq. (5).
8:      Compute distill loss $L_{distill}$.
9:      Backward $L_{Aux}$ and $L_{distill}$.
10: **until** Student model converges

---

The MPO pseudocode is shown in Algorithm S.2.

---

**Algorithm S.2** MPO decomposition for a matrix.

---

**Input:** matrix $\mathbf{M}$, the number of local tensors $n$
**Output** : MPO tensor list $\{\mathcal{T}_{(k)}\}_{k=1}^{n}$

1:  **for** $k = 1 \rightarrow n - 1$ **do**
2:      $\mathbf{M}[I, J] \longrightarrow \mathbf{M}[d_{k-1} \times i_k \times j_k, -1]$
3:      $\mathbf{U}\lambda\mathbf{V}^\top = \text{SVD}\,(\mathbf{M})$
4:      $\mathbf{U}[d_{k-1} \times i_k \times j_k, d_k] \longrightarrow \mathcal{U}[d_{k-1}, i_k, j_k, d_k]$
5:      $\mathcal{T}^{(k)} := \mathcal{U}$
6:      $\mathbf{M} := \lambda\mathbf{V}^\top$
7:  **end for**
8:  $\mathcal{T}^{(n)} := \mathbf{M}$
9:  Normalization
10: **return** $\{\mathcal{T}_{(k)}\}_{k=1}^{n}$

---

## D    Addition Experiment Results

### D.1    Memory and time cost

The distillation cost (memory and time cost) of the original model and the model after applying OPDF are shown in Table S.1. We can observe that as the number of parameters obtained from MPO decomposition increases, both the training time and memory cost increase. However, as the dataset size increases, the ratio of additional time and memory required for training by OPDF to the original training requirements generally exhibits a decreasing trend (e.g., 0.6/0.4 for RTE vs 0.3/0.1 for MNLI in BERT-of-Theseus model). Therefore, the additional time and memory introduced by our method become less of a critical bottleneck affecting the training speed as the dataset size increases.

Table S.1: Training time and Memory Cost. (Train time(S) / Memory Cost(GB))

| Datasets | RTE | MRPC | STS-B | CoLA | SST-2 | QNLI | Q QP | MNLI |
|---|---|---|---|---|---|---|---|---|
| **BERT-of-Theseus** | | | | | | | | |
| None | 400.2/14.8 | 739.5/14.8 | 754.4/14.8 | 1365.4/14.8 | 2553.7/14.1 | 3514.2/14.1 | 7518.7/14.1 | 8873.9/13.5 |
| +OPDF (Ours) | 625.8/20.7 | 1054.9/20.6 | 1932.0/20.5 | 2560.4/16.8 | 21864.5/22.1 | 5041.6/27.0 | 10301.9/18.9 | 11674.8/14.8 |
| **LGTM** | | | | | | | | |
| None | 1086.1/9.7 | 1408.5/9.6 | 2049.1/9.6 | 3358.9/9.6 | 5270.9/9.6 | 8142.8/9.6 | 30272.6/9.6 | 31554.8/9.6 |
| +OPDF (Ours) | 1976.3/19.7 | 2611.9/19.6 | 3603.6/19.7 | 2348.4/19.6 | 9983.4/19.6 | 14838.4/19.6 | 44058.5/14.7 | 47849.4/19.6 |
| **DBKD** | | | | | | | | |
| None | 40.7/2.1 | 80.3/3.2 | NA | 186.4/3.2 | 1355.8/3.2 | 2149.2/3.2 | 7487.5/3.2 | 15513.6/3.2 |
| +OPDF (Ours) | 93.1/5.0 | 213.0/6.6 | NA | 373.3/6.2 | 2793.8/5.4 | 6076.5/6.6 | 14030.7/6.6 | 21273.4/5.0 |
| **AD-KD** | | | | | | | | |
| None | 308.5/3.8 | 351.3/3.8 | 495.6/3.8 | 780.3/5.9 | 3637.4/5.9 | 5832.7/5.9 | 28763.3/5.9 | 41898.4/20.6 |
| +OPDF (Ours) | 1156.8/14.4 | 1391.3/14.5 | 1604.2/12.1 | 2249.8/18.1 | 8802.3/14.5 | 13551.1/14.5 | 63695.5/14.5 | 65735.8/34.0 |

We show the time of overparameterization using MPO and the contraction of decomposed matrices into the original matrix in Table S.2 as follows. It can be observed that the time required for decomposition and reconstruction is acceptable compared to the training duration.

Table S.2: The spending time (s) of decomposing and reconstructing.

| Datasets | RTE | MRPC | STS-B | CoLA | SST-2 | QNLI | QQP | MNLI |
|---|---|---|---|---|---|---|---|---|
| **BERT-of-Theseus** | | | | | | | | |
| Decompose | 397.4 | 308.5 | 400.2 | 154.6 | 797.7 | 671.6 | 584.7 | 137.5 |
| Reconstruct | 2.3 | 2.0 | 2.4 | 0.7 | 12.8 | 3.3 | 10.9 | 0.8 |
| **LGTM** | | | | | | | | |
| Decompose | 403.6 | 369.5 | 377.8 | 192.2 | 131.4 | 123.9 | 80.2 | 83.0 |
| Reconstruct | 8.6 | 6.8 | 7.2 | 2.6 | 1.0 | 0.9 | 1.0 | 1.5 |
| **DBKD** | | | | | | | | |
| Decompose | 117.5 | 189.1 | NA | 168.5 | 153.1 | 166.2 | 110.7 | 165.0 |
| Reconstruct | 0.9 | 1.0 | NA | 0.8 | 0.9 | 0.8 | 0.7 | 0.8 |
| **AD-KD** | | | | | | | | |
| Decompose | 291.7 | 313.0 | 232.6 | 235.5 | 119.0 | 148.0 | 148.1 | 171.6 |
| Reconstruct | 1.6 | 1.8 | 1.2 | 1.4 | 0.9 | 1.0 | 2.5 | 1.2 |

## D.2 Experimental Details

As illustrated in Eq. (2), when a parameter matrix $\mathbf{W}$ is given, its MPO decomposition into a product of $n$ local tensors can be represented as follows:

$$\text{MPO}\,(\mathbf{W}) = \mathcal{T}^{i_1,i_2,i_3,\ldots,i_n}_{j_1,j_2,j_3,\ldots,j_n}. \tag{S.10}$$

The models mentioned—Bert of theseus [56], LGTM [44], DBKD [45] and AD-KD [46]—are all variants of BERT, which itself is built using transformer blocks. We decompose both the feed-forward network and the multi-head attention layer within the transformer block. Moreover, the teacher model must employ a decomposition granularity that is consistent with that of the student model to ensure proper alignment of auxiliary tensors. When calculating the auxiliary loss $\mathcal{L}_{Aux}$, the alignment is typically between the n-th layer of the student model and the N-th layer of the teacher model, where N is generally an integer multiple of n. The detailed hyperparameter settings for these NLP distillation models are provided in Table S.3 and S.4. In NLP tasks, our method takes half to two GPU hours on A100 GPU.

Additionally, we implement OPDF on TinyViT [61] to demonstrate its applicability as an orthogonal approach across various knowledge distillation methods that utilize the transformer architecture. Unlike NLP models, we decompose the projection layer in addition to the feed-forward network and multi-head attention layer in the vision transformer block. The specific experimental parameters utilized are detailed in Table S.5. In CV tasks, our method takes 160.0 GPU days on A100 GPUs to pretrain TinyViT-21M. We report the performance of the model that achieves the best results on the validation set when applied to the test set.

Table S.3: The feed-forward network layer settings in NLP distilation model.

| Experiments | RTE | MRPC | STS-B | CoLA |
|---|---|---|---|---|
| **BERT-of-Theseus [56]** | | | | |
| SVD | $\mathcal{T}^{32,24}_{64,48}$ | $\mathcal{T}^{32,24}_{64,48}$ | $\mathcal{T}^{32,24}_{64,48}$ | $\mathcal{T}^{32,24}_{64,48}$ |
| OPDF (Ours) | $\mathcal{T}^{32,1,1,1,24}_{64,1,1,1,48}$ | $\mathcal{T}^{32,1,1,1,24}_{64,1,1,1,48}$ | $\mathcal{T}^{32,1,1,1,24}_{64,1,1,1,48}$ | $\mathcal{T}^{32,1,24}_{64,1,48}$ |
| **LGTM [44]** | | | | |
| SVD | $\mathcal{T}^{32,24}_{64,48}$ | $\mathcal{T}^{32,24}_{64,48}$ | $\mathcal{T}^{32,24}_{64,48}$ | $\mathcal{T}^{32,24}_{64,48}$ |
| OPDF (Ours) | $\mathcal{T}^{32,1,1,1,24}_{64,1,1,1,48}$ | $\mathcal{T}^{32,1,1,1,24}_{64,1,1,1,48}$ | $\mathcal{T}^{32,1,1,1,24}_{64,1,1,1,48}$ | $\mathcal{T}^{32,1,1,1,1,24}_{64,1,1,1,1,48}$ |
| **DBKD [45]** | | | | |
| SVD | $\mathcal{T}^{32,24}_{64,48}$ | $\mathcal{T}^{32,24}_{64,48}$ | / | $\mathcal{T}^{32,24}_{64,48}$ |
| OPDF (Ours) | $\mathcal{T}^{32,1,1,1,1,24}_{64,1,1,1,1,48}$ | $\mathcal{T}^{32,1,1,1,1,1,1,1,24}_{64,1,1,1,1,1,1,1,48}$ | / | $\mathcal{T}^{32,1,1,1,1,1,1,1,24}_{64,1,1,1,1,1,1,1,48}$ |
| **AD-KD [46]** | | | | |
| SVD | $\mathcal{T}^{32,24}_{64,48}$ | $\mathcal{T}^{32,24}_{64,48}$ | $\mathcal{T}^{32,24}_{64,48}$ | $\mathcal{T}^{32,24}_{64,48}$ |
| OPDF (Ours) | $\mathcal{T}^{32,1,1,1,1,1,1,1,24}_{64,1,1,1,1,1,1,1,48}$ | $\mathcal{T}^{32,1,1,1,24}_{64,1,1,1,48}$ | $\mathcal{T}^{32,1,1,1,1,1,24}_{64,1,1,1,1,1,48}$ | $\mathcal{T}^{32,1,24}_{64,1,48}$ |

| Experiments | SST-2 | QNLI | QQP | MNLI |
|---|---|---|---|---|
| **BERT-of-Theseus** | | | | |
| SVD | $\mathcal{T}^{32,24}_{64,48}$ | $\mathcal{T}^{32,24}_{64,48}$ | $\mathcal{T}^{32,24}_{64,48}$ | $\mathcal{T}^{32,24}_{64,48}$ |
| OPDF (Ours) | $\mathcal{T}^{32,1,1,1,1,24}_{64,1,1,1,1,48}$ | $\mathcal{T}^{32,1,1,1,1,1,1,1,24}_{64,1,1,1,1,1,1,1,48}$ | $\mathcal{T}^{32,1,1,1,24}_{64,1,1,1,48}$ | $\mathcal{T}^{32,1,24}_{64,1,48}$ |
| **LGTM** | | | | |
| SVD | $\mathcal{T}^{32,24}_{64,48}$ | $\mathcal{T}^{32,24}_{64,48}$ | $\mathcal{T}^{32,24}_{64,48}$ | $\mathcal{T}^{32,24}_{64,48}$ |
| OPDF (Ours) | $\mathcal{T}^{32,1,1,1,24}_{64,1,1,1,48}$ | $\mathcal{T}^{32,1,1,1,24}_{64,1,1,1,48}$ | $\mathcal{T}^{32,1,24}_{64,1,48}$ | $\mathcal{T}^{32,1,1,1,24}_{64,1,1,1,48}$ |
| **DBKD** | | | | |
| SVD | $\mathcal{T}^{32,24}_{64,48}$ | $\mathcal{T}^{32,24}_{64,48}$ | $\mathcal{T}^{32,24}_{64,48}$ | $\mathcal{T}^{32,24}_{64,48}$ |
| OPDF (Ours) | $\mathcal{T}^{32,1,1,1,1,24}_{64,1,1,1,1,48}$ | $\mathcal{T}^{32,1,1,1,1,1,1,1,24}_{64,1,1,1,1,1,1,1,48}$ | $\mathcal{T}^{32,1,1,1,24}_{64,1,1,1,48}$ | $\mathcal{T}^{32,1,1,1,1,1,1,1,24}_{64,1,1,1,1,1,1,1,48}$ |
| **AD-KD** | | | | |
| SVD | $\mathcal{T}^{32,24}_{64,48}$ | $\mathcal{T}^{32,24}_{64,48}$ | $\mathcal{T}^{32,24}_{64,48}$ | $\mathcal{T}^{32,24}_{64,48}$ |
| OPDF (Ours) | $\mathcal{T}^{32,1,1,24}_{64,1,1,48}$ | $\mathcal{T}^{32,1,1,24}_{64,1,1,48}$ | $\mathcal{T}^{32,1,1,24}_{64,1,1,48}$ | $\mathcal{T}^{32,1,1,1,24}_{64,1,1,1,48}$ |

Table S.4: The multi-head attention layer settings in NLP distilation model.

| Experiments | RTE | MRPC | STS-B | CoLA |
|---|---|---|---|---|
| **BERT-of-Theseus** | | | | |
| SVD | $\mathcal{T}^{32,24}_{32,24}$ | $\mathcal{T}^{32,24}_{32,24}$ | $\mathcal{T}^{32,24}_{32,24}$ | $\mathcal{T}^{32,24}_{32,24}$ |
| OPDF (Ours) | $\mathcal{T}^{32,1,1,1,24}_{32,1,1,1,24}$ | $\mathcal{T}^{32,1,1,1,24}_{32,1,1,1,24}$ | $\mathcal{T}^{32,1,1,1,24}_{32,1,1,1,24}$ | $\mathcal{T}^{32,1,24}_{32,1,24}$ |
| **LGTM** | | | | |
| SVD | $\mathcal{T}^{32,24}_{32,24}$ | $\mathcal{T}^{32,24}_{32,24}$ | $\mathcal{T}^{32,24}_{32,24}$ | $\mathcal{T}^{32,24}_{32,24}$ |
| OPDF (Ours) | $\mathcal{T}^{32,1,1,1,24}_{32,1,1,1,24}$ | $\mathcal{T}^{32,1,1,1,24}_{32,1,1,1,24}$ | $\mathcal{T}^{32,1,1,1,24}_{32,1,1,1,24}$ | $\mathcal{T}^{32,1,1,1,1,24}_{32,1,1,1,1,24}$ |
| **DBKD** | | | | |
| SVD | $\mathcal{T}^{32,24}_{32,24}$ | $\mathcal{T}^{32,24}_{32,24}$ | / | $\mathcal{T}^{32,24}_{32,24}$ |
| OPDF (Ours) | $\mathcal{T}^{32,1,1,1,24}_{32,1,1,1,24}$ | $\mathcal{T}^{32,1,1,1,1,1,1,24}_{32,1,1,1,1,1,1,24}$ | / | $\mathcal{T}^{32,1,1,1,1,1,1,24}_{32,1,1,1,1,1,1,24}$ |
| **AD-KD** | | | | |
| SVD | $\mathcal{T}^{32,24}_{32,24}$ | $\mathcal{T}^{32,24}_{32,24}$ | $\mathcal{T}^{32,24}_{32,24}$ | $\mathcal{T}^{32,24}_{32,24}$ |
| OPDF (Ours) | $\mathcal{T}^{32,1,1,1,1,1,1,1,24}_{32,1,1,1,1,1,1,1,24}$ | $\mathcal{T}^{32,1,1,1,24}_{32,1,1,1,24}$ | $\mathcal{T}^{32,1,1,1,1,1,24}_{32,1,1,1,1,1,24}$ | $\mathcal{T}^{32,1,24}_{32,1,24}$ |

| Experiments | SST-2 | QNLI | QQP | MNLI |
|---|---|---|---|---|
| **BERT-of-Theseus** | | | | |
| SVD | $\mathcal{T}^{32,24}_{32,24}$ | $\mathcal{T}^{32,24}_{32,24}$ | $\mathcal{T}^{32,24}_{32,24}$ | $\mathcal{T}^{32,24}_{32,24}$ |
| OPDF (Ours) | $\mathcal{T}^{32,1,1,1,1,24}_{32,1,1,1,1,24}$ | $\mathcal{T}^{32,1,1,1,1,1,1,1,24}_{32,1,1,1,1,1,1,1,24}$ | $\mathcal{T}^{32,1,1,1,24}_{32,1,1,1,24}$ | $\mathcal{T}^{32,1,24}_{32,1,24}$ |
| **LGTM** | | | | |
| SVD | $\mathcal{T}^{32,24}_{32,24}$ | $\mathcal{T}^{32,24}_{32,24}$ | $\mathcal{T}^{32,24}_{32,24}$ | $\mathcal{T}^{32,24}_{32,24}$ |
| OPDF (Ours) | $\mathcal{T}^{32,1,1,1,24}_{32,1,1,1,24}$ | $\mathcal{T}^{32,1,1,1,24}_{32,1,1,1,24}$ | $\mathcal{T}^{32,1,24}_{32,1,24}$ | $\mathcal{T}^{32,1,1,1,24}_{32,1,1,1,24}$ |
| **DBKD** | | | | |
| SVD | $\mathcal{T}^{32,24}_{32,24}$ | $\mathcal{T}^{32,24}_{32,24}$ | $\mathcal{T}^{32,24}_{32,24}$ | $\mathcal{T}^{32,24}_{32,24}$ |
| OPDF (Ours) | $\mathcal{T}^{32,1,1,1,1,1,24}_{32,1,1,1,1,1,24}$ | $\mathcal{T}^{32,1,1,1,1,1,1,1,24}_{32,1,1,1,1,1,1,1,24}$ | $\mathcal{T}^{32,1,1,1,24}_{32,1,1,1,24}$ | $\mathcal{T}^{32,1,1,1,1,1,1,1,24}_{32,1,1,1,1,1,1,1,24}$ |
| **AD-KD** | | | | |
| SVD | $\mathcal{T}^{32,24}_{32,24}$ | $\mathcal{T}^{32,24}_{32,24}$ | $\mathcal{T}^{32,24}_{32,24}$ | $\mathcal{T}^{32,24}_{32,24}$ |
| OPDF (Ours) | $\mathcal{T}^{32,1,1,1,24}_{32,1,1,1,24}$ | $\mathcal{T}^{32,1,1,1,24}_{32,1,1,1,24}$ | $\mathcal{T}^{32,1,1,1,24}_{32,1,1,1,24}$ | $\mathcal{T}^{32,1,1,1,24}_{32,1,1,1,24}$ |

Table S.5: The experiment settings in CV distilation model.

| Experiments | Feed-forward Network | | | Multi-head Attention | | | Projection Layer | | |
|---|---|---|---|---|---|---|---|---|---|
| | layer1 | layer2 | layer3 | layer1 | layer2 | layer3 | layer1 | layer2 | layer3 |
| **TinyVit-5M** | | | | | | | | | |
| SVD | $\mathcal{T}^{16,8}_{32,16}$ | $\mathcal{T}^{16,10}_{32,20}$ | $\mathcal{T}^{20,16}_{40,32}$ | $\mathcal{T}^{16,8}_{24,16}$ | $\mathcal{T}^{16,10}_{24,20}$ | $\mathcal{T}^{20,16}_{32,30}$ | $\mathcal{T}^{16,8}_{16,8}$ | $\mathcal{T}^{16,10}_{16,10}$ | $\mathcal{T}^{20,16}_{20,16}$ |
| OPDF (Ours) | $\mathcal{T}^{16,1,8}_{32,1,16}$ | $\mathcal{T}^{16,1,10}_{32,1,20}$ | $\mathcal{T}^{20,1,16}_{40,1,32}$ | $\mathcal{T}^{16,8}_{24,1,16}$ | $\mathcal{T}^{16,1,10}_{24,1,20}$ | $\mathcal{T}^{20,1,16}_{32,1,30}$ | $\mathcal{T}^{16,1,8}_{16,1,8}$ | $\mathcal{T}^{16,1,10}_{16,1,10}$ | $\mathcal{T}^{20,1,16}_{20,1,16}$ |
| **TinyVit-11M** | | | | | | | | | |
| SVD | $\mathcal{T}^{16,8}_{32,16}$ | $\mathcal{T}^{16,16}_{32,32}$ | $\mathcal{T}^{32,14}_{56,32}$ | $\mathcal{T}^{16,8}_{24,16}$ | $\mathcal{T}^{16,16}_{32,24}$ | $\mathcal{T}^{32,14}_{48,28}$ | $\mathcal{T}^{16,8}_{16,8}$ | $\mathcal{T}^{16,16}_{16,16}$ | $\mathcal{T}^{32,14}_{32,14}$ |
| OPDF (Ours) | $\mathcal{T}^{16,1,8}_{32,1,16}$ | $\mathcal{T}^{16,1,16}_{32,1,32}$ | $\mathcal{T}^{32,1,14}_{56,1,32}$ | $\mathcal{T}^{16,1,8}_{24,1,16}$ | $\mathcal{T}^{16,1,16}_{32,1,24}$ | $\mathcal{T}^{32,1,14}_{48,1,28}$ | $\mathcal{T}^{16,1,8}_{16,1,8}$ | $\mathcal{T}^{16,1,16}_{16,1,16}$ | $\mathcal{T}^{32,1,14}_{32,1,14}$ |
| **TinyVit-21M** | | | | | | | | | |
| SVD | $\mathcal{T}^{24,8}_{32,24}$ | $\mathcal{T}^{24,16}_{48,32}$ | $\mathcal{T}^{32,18}_{64,36}$ | $\mathcal{T}^{24,8}_{32,18}$ | $\mathcal{T}^{24,16}_{36,32}$ | $\mathcal{T}^{32,18}_{54,32}$ | $\mathcal{T}^{24,8}_{24,8}$ | $\mathcal{T}^{24,16}_{24,16}$ | $\mathcal{T}^{32,18}_{32,18}$ |
| OPDF (Ours) | $\mathcal{T}^{24,1,8}_{32,1,24}$ | $\mathcal{T}^{24,1,16}_{48,1,32}$ | $\mathcal{T}^{32,1,18}_{64,1,36}$ | $\mathcal{T}^{24,1,8}_{32,1,18}$ | $\mathcal{T}^{24,1,16}_{36,1,32}$ | $\mathcal{T}^{32,1,18}_{54,1,32}$ | $\mathcal{T}^{24,1,8}_{24,1,8}$ | $\mathcal{T}^{24,1,16}_{24,1,16}$ | $\mathcal{T}^{32,1,18}_{32,1,18}$ |

