# OpenReview forum: "Over-parameterized Student Model via Tensor Decomposition Boosted Knowledge Distillation"
_NeurIPS.cc/2024/Conference — NeurIPS 2024 poster_

### Official Review · Reviewer_XV5e · 2024-06-17

**Soundness:** 3
**Presentation:** 3
**Contribution:** 3
**Rating:** 5
**Confidence:** 4

**Summary:**

In this work, the authors focus on the knowledge distillation (KD) task, using tensor decomposition to enhance the performance of the student model. Leveraging the principle of overparameterization, the authors employ the Matrix Product Operator (MPO), also known as tensor train matrix, to reformulate the original weight matrix. Additionally, they propose a “distillation loss” to measure the distance between the student’s weights and the teacher’s weights. In the experiments, the proposed method is integrated with existing KD methods, and the results demonstrate a clear improvement in KD performance.

**Strengths:**

The paper introduces an innovative application of tensor decomposition. Typically, tensor decomposition is used for dimension reduction, adhering to the low-rank principle. However, this paper utilizes tensor decomposition in a novel manner: employing MPO to construct overparameterized learning models. While I am not entirely convinced why the overparameterized MPO is superior to traditional matrix decomposition or other tensor networks, the numerical results indicate that this approach could be a promising new avenue for using tensor networks to solve more machine learning problems.

**Weaknesses:**

The clarity of the paper needs improvement. For instance, Figure 1 is not fully comprehensible as the meaning of the arrows is not clearly explained. Additionally, the experiment settings do not clearly describe how the weights are reshaped into the higher-order tensor format.

**Questions:**

1.	Since the entire paper is based on the principle of overparameterization of the student model, it would be beneficial to explain in the preliminary section why the overparameterization principle is relevant to the KD problem. This addition would help non-experts follow the main idea of the paper more smoothly.

2.	In lines 176-177, the phrase “losing its ability to think independently” is highlighted. I am confused by this statement. Could you provide more interpretation of this claim, supported with sufficient evidence?

3.	Please offer more interpretation of the central and auxiliary tensors mentioned in lines 184-185, using formulas or figures, for example. I cannot clearly understand their differences from the current non-rigorous descriptions.

4.	Why does using MPO provide better performance than SVD? Is it because the MPO forms more “linear layers” than SVD? Have you considered other tensor network formats such as tree tensor networks?

5.	How does the selection of TT-ranks (e.g., d_k in Eq. 2) affect the performance?

**Limitations:**

The paper lacks a discussion on the limitations of the proposed method. I suggest that the authors address this aspect to provide a more balanced and comprehensive evaluation of their work.

---

> ### Author Rebuttal · Authors · 2024-08-03
>
> We sincerely thank you for the constructive comments and suggestions, which are very helpful in improving our paper. The following responses will be incorporated into the revised paper.
>
> **Q1. The impact of over-parameterization on student model performance.**
>
> **Reply:** Thank for your excellent comment. Since the performance of student models are typically limited by their number of parameters, increasing the number of model parameters can significantly enhance performance (*e.g.* TinyViT-21M (91.9) *v.s.* TinyViT-5M (87.6) in top-5 accuracy on Imagenet V2). Thus, increasing parameters is beneficial for enhancing the performance of the student model. We will include information about the impact of over-parameterization on the performance of student model in the revised version of the preliminary section.
>
> **Q2. The meaning of "think independantly".**
>
> **Reply:** Great remark! We will rephrase this statement to be "learn independently". In the OPDF, auxiliary tensor alignment allows the student model to learn task-relevant information. During the distillation process, learning through the central tensor not only enables the model to imitate the teacher model, but also equips it with the capability to learn independently from the original labels, detached from the teacher model. Consequently, this endows the model with the potential to surpass the teacher model.
>
> Conversely, if the entire parameter matrix is directly aligned, then the student model can only emulate the teacher model. The effectiveness of this method is further evidenced by the experimental results. As shown in Table 1 in our paper, after incorporating OPDF, the student model outperforms the teacher model by 1.2% on the RTE dataset.
>
> **Q3. The interpretation of central and auxiliary tensors.**
>
> **Reply:** Thanks for your comment. The definition of central and auxiliary tensors is shown in lines 152-153 in our paper (please kindly refer to the Figure 1b). MPO allows for the decomposition of parameter matrices into a series of tensors:
>
> $${MPO}~(\mathbf{M})= \prod_{k=1}^n \mathbf{T_{(k)}}[d_{k-1},i_k,j_k,d_k].\tag{1}$$
>
> $$d_k = \min\bigg(\prod_{m=1}^k i_m\times j_m, \prod_{m=k+1}^n i_m\times j_m\bigg).\tag{2}$$
>
> Following Refs. [1,2], the tensor right in the middle is termed as central tensor, and the rest as auxiliary tensor.
>
> *References:*
>
> [1] P. Liu, et al. Enabling Lightweight Fine-tuning for Pre-trained Language Model Compression based on Matrix Product Operators.ACL2021
>
> [2] Z.F. Gao, et al. Parameter-Efficient Mixture-of-Experts Architecture for Pre-trained Language Models. COLING2022.
>
>
> **Q4. Reasons why MPO outperforms SVD  and the justification for not using MPO over other tensor networks.**
>
> **Reply:** This is an insightful question. MPO generally surpasses SVD because adding eigenvalue dimensions in SVD improves very little of the model capacity, due to the fixed degrees of freedom, expanding parameters through SVD has an upper limit. In contrast, MPO employs matrices of eigenvectors without increasing eigenvalue dimensions, enabling potentially unlimited growth in model parameters.
>
> We also have considered other tensor network format like CP decomposition (CPD) and Tucker decomposition. Generally, the algorithm capacity is larger as the number of tensors $n$ increases (denoting more tensors). When $n > 3$, MPO has smaller time complexity than Tucker decomposition. It noted that SVD can be considered as a special case of MPO when $n = 2$ and CPD is a special case of Tucker when the core tensor is the super-diagonal matrix.  The specifics are illustrated in **Table A**. We will investigate other tensor networks as part of our further work.
>
> **Table A.** Inference time complexities of different low-rank approximation methods. Here, $n$ denotes the number of the tensors, $m$ denotes $max({{{\{i_k\} }^n_{k=1}}})$ means the largest ${i_k}$ in input list, and $d$ denotes $max({{{ \{d_k^{'} \} }^n_{k=0}}})$  meaning the largest dimension $d_{k}^{'}$ in the truncated dimension list.
>
> | Category    | Method| Inference Time |
> | ----- | :-------------------: | :--------------: |
> | **Trucker** | $Trucker_{(d=1)} (CP)$ | $O(nmd^2)$     |
> | **Trucker** | $Trucker_{(d>1)}$          | $O(nmd + d^n)$  |
> | **MPO**     | $MPO_{(n=2)} (SVD)$    | $O(2md^3)$     |
> | **MPO**     | $MPO_{(n>2)}$              | $O(nmd^3)$     |
>
>
> **Q5. The impact of TT-Ranks on performance.**
>
> **Reply:** Excellent comment! Indeed, we have conducted experiments regarding the impact of $d_k$ on the model performance (please kindly refer to Appendix Table S.4). We can observe that the performance of our approach consistently stabilizes around certain values, indicating that our method is *not sensitive* to the specific MPO techniques used. Therefore, when over-parameterizing, we should focus primarily on the decomposition scale rather than the MPO method employed.
>
> **Q6. The limitations of OPDF.**
>
> **Reply:** Great remark! As discussed in Section 5.3 in our paper, while OPDF can enhance the performance of the student model through over-parameterization, there are inherent limits to these benefits. Additionally, due to over-parameterizing the KD model, OPDF results in higher memory consumption and longer training cost. The memory usage and training cost before and after using OPDF can be found in **Table 1** in the *one-page PDF rebuttal file*.
>
> While OPDF increases the memory usage and training time, this impact diminishes as the dataset size grows, meaning that the ratio of additional memory and training time to the original requirements decreases. We will include an enhanced discussion on the limitations of OPDF in the revised paper.
>
> **Concluding remark:** We sincerely thank you for putting forward excellent comments. We hope the above responses are helpful to clarify your questions. We look forward to addressing any additional questions. Your consideration of improving the rating of our paper will be much appreciated!

---

> ### Author Response · Authors · 2024-08-11
> **Looking forward to your feedback**
>
> Dear Reviewer XV5e,
>
> We're keen to know if there are any remaining concerns that require attention or if there are additional discussions that should take place. Your insights are greatly appreciated as they contribute to the refinement of the paper. Looking forward to your feedback. Thank you.
>
> Best regards,
>
> The Authors

---

> ### Author Response · Authors · 2024-08-12
> **Request your feedback before the end of the discussion period**
>
> Dear Reviewer XV5e:
>
> As the author-reviewer discussion period will end soon, we would appreciate it if you could kindly review our responses at your earliest convenience. If there are any further questions or comments, we will do our best to address them before the discussion period ends.
>
> Thank you very much for your time and efforts!
>
> Sincerely,
>
> The Authors

---

> ### Author Response · Authors · 2024-08-13
> **Kindly request your feedback before the end of the discussion period**
>
> Dear Reviewer XV5e:
>
> As the author-reviewer discussion period is soon ending, we would appreciate it if you could review our responses and provide your feedback at your earliest convenience. If there are any further questions or comments, we will do our best to address them before the discussion period ends.
>
> Thank you very much for your time and efforts!
>
> Sincerely,
>
> The Authors

---

> > ### Comment · Reviewer_XV5e · 2024-08-13
> >
> > Sorry for the delayed reply. I appreciated the detailed and clear  response from the side of authors. It answered most of my concern in the review. I would adjust the recommendation score from 4 to 5.

---

> > > ### Author Response · Authors · 2024-08-13
> > > **Thank you for raising the score**
> > >
> > > Dear XV5e:
> > >
> > > Thank you for your positive feedback. We will include the additional experiments and texts in the revised paper.
> > >
> > > Thank you for your time and efforts!
> > >
> > > Best regards,
> > >
> > > The Authors

---

### Official Review · Reviewer_Dajm · 2024-06-30

**Soundness:** 4
**Presentation:** 4
**Contribution:** 3
**Rating:** 7
**Confidence:** 4

**Summary:**

This paper introduces the Over-Parameterization Distillation Framework (OPDF), which addresses performance degradation in limited-parameter student networks after knowledge distillation (KD). OPDF proposes an overparameterized student model that utilizes the tensor-decomposition technique known as matrix product operator (MPO), allowing for a significant increase in parameters during student training time without imposing additional inference time burden. Experimental validation is performed across multiple KD tasks to assess the effectiveness of the proposed technique.

**Strengths:**

+ The proposed technique appears to be quite novel. The use of MPO to expand parameters in the student model during the KD process, along with the tensor alignment loss function to improve student model performance, introduces innovative approaches that could offer significant advantages, particularly on low-computational devices.
+ The proposed methodology is easy to understand, even though it includes some abstract concepts. The authors have effectively structured their methodology by first providing a high-level overview of their technique with an illustrative figure, followed by detailed explanations of their important components.
+ Extensive experiments are conducted across both NLP and CV tasks. Multiple knowledge distillation (KD) techniques are employed to demonstrate how their  contributions is orthogonal to existing methods.   Also, parameters introduced in training as well as inference process are clearly shown.
+ The study extensively examines the impact of overparameterization scale, learning rate, and other components of ODPF through ablation experiments.  This study helps to justify the effectiveness of their technique.

**Weaknesses:**

- The paper does not provide information on the time required for the student network's overparameterization using MPO and the contraction  of decomposed matrices into the original matrix. Having this information would be important for understanding the practical implications of the technique.
- The experiments are conducted on a relatively smaller model. I am curious about the feasibility of applying this technique to LLM/VLMs with billions of parameters. A significant concern is whether their approach, which involves decomposing the teacher network, can scale effectively to such large models. A deeper discussion on this topic would provide valuable insights.

**Questions:**

Please refer to Weaknesses section

---

> ### Author Rebuttal · Authors · 2024-08-03
>
> We sincerely thank you for the positive feedback along with constructive comments and suggestions, which are very helpful in improving our paper. We are also grateful that you recognized the strengths and contributions of our paper. Moreover, the following responses will be incorporated into the revised paper.
>
>
> **Q1. Time Consumption for decomposing and reconstructing the parameter matrix.**
>
> **Reply:** This is a great remark. We listed the time of overparameterization using MPO and the contraction of decomposed matrices into the original matrix in **Table A** as follows. It can be observed that the time required for decomposition and reconstruction is acceptable compared to the training duration (please alse see Appendix C).
>
> **Table A.** The spending time (s) of decomposition and reconstruction.
>
> | Cases                             | RTE   | MRPC  | STS-B | CoLA  | SST-2 | QNLI  | QQP   | MNLI  |
> | ------------------------------- | :-----: | :-----: | :-----: | :-----: | :-----: | :-----: | :-----: | :-----: |
> | **BERT-of-Theseus** Decomposition   | 397.4 | 308.5 | 400.2 | 154.6 | 797.7 | 671.6 | 584.7 | 137.5 |
> | **BERT-of-Theseus** Reconstruction | 2.3   | 2.0   | 2.4   | 0.7   | 12.8  | 3.3   | 10.9  | 0.8   |
> | **LGTM** Decomposition              | 403.6 | 369.5 | 377.8 | 192.2 | 131.4 | 123.9 | 80.2  | 83.0  |
> | **LGTM** Reconstruction           | 8.6   | 6.8   | 7.2   | 2.6   | 1.6   | 0.9   | 1.0   | 1.5   |
> | **DBKD** Decomposition              | 117.5 | 189.1 | Na     | 168.5 | 153.1 | 166.2 | 110.7 | 165.0 |
> | **DBKD** Reconstruction            | 0.9   | 1.0   | Na     | 0.8   | 0.9   | 0.8   | 0.7   | 0.8   |
> | **AD-KD** Decomposition             | 291.7 | 313.0 | 232.6 | 235.5 | 119.0 | 148.0 | 148.1 | 171.6 |
> | **AD-KD** Reconstruction          | 1.6   | 1.8   | 1.2   | 1.4   | 0.9   | 1.0   | 2.5   | 1.2   |
>
>
>
> **Q2. Applying OPDF to model with billions of parameters.**
>
> **Reply:** This is an insightful question. We have implemented OPDF on the GPT-2-760M, OPT-6.7B, and LLAMA-7B models, with corresponding teacher models of GPT-2-1.5B, OPT-13B, and LLAMA-13B, respectively. The Rouge-L scores of these models on five instruction-following datasets are presented in **Table B** (see below). Our results indicate that OPDF significantly enhances the distillation efficiency for larger models across all datasets, demonstrating its efficacy even for models with billions of parameters.
>
> **Table B.** Distillation results on larger models with OPDF.
>
> | Model     | #Params | Method  | Dolly    | SelfInst | Vicuna   | S-NI     | UnNI     | Avg.     | # Train Params | # Inference Params |
> | --------- | :-------: | :-------: | :--------: | :--------: | :--------: | :--------: | :--------: | :--------: | :--------------: | :------------------: |
> | **GPT-2** | 1.5B    | Teacher | 27.6     | 14.3     | 16.3     | 27.6     | 31.8     | 23.5     | 1.5B           | 1.5B               |
> | **GPT-2** | 760M    | w/o KD  | 25.4     | 12.4     | 16.1     | 21.5     | 24.0     | 19.9     | 760M           | 760M               |
> | **GPT-2** | 760M    | KD      | 25.9     | 13.4     | 16.9     | 25.3     | 28.0     | 21.9     | 760M           | 760M               |
> | **GPT-2** | 760M    | KD+OPDF | **26.1** | **14.1** | **17.5** | **25.7** | **28.6** | **22.4** | 1.3B           | 760M               |
> | **OPT**   | 13B     | Teacher | 29.2     | 18.4     | 17.8     | 30.4     | 36.1     | 26.4     | 13B            | 13B                |
> | **OPT**   | 6.7B    | w/o KD  | 27.6     | 16.4     | 17.8     | 30.3     | 28.6     | 24.1     | 6.7B           | 6.7B               |
> | **OPT**   | 6.7B    | KD      | 28.3     | 17.0     | 17.5     | 30.7     | 26.7     | 24.0     | 6.7B           | 6.7B               |
> | **OPT**   | 6.7B    | KD+OPDF | **28.5** | **17.7** | **17.9** | **31.4** | **29.8** | **25.1** | 14B            | 6.7B               |
> | **LLaMA** | 13B     | Teacher | 29.7     | 23.4     | 19.4     | 35.8     | 38.5     | 29.4     | 13B            | 13B                |
> | **LLaMA** | 7B      | w/o KD  | 26.3     | 20.8     | 17.5     | 32.4     | 35.8     | 26.6     | 7B             | 7B                 |
> | **LLaMA** | 7B      | KD      | 27.4     | 20.2     | 18.4     | 33.7     | 37.9     | 27.5     | 7B             | 7B                 |
> | **LLaMA** | 7B      | KD+OPDF | **27.5** | **21.6** | **19.7** | **34.8** | **40.0** | **28.7** | 10B            | 7B                 |

---

> > ### Comment · Reviewer_Dajm · 2024-08-13
> > **Thanks for the Rebuttal**
> >
> > Dear Authors,
> > Thank you for your effort toward addressing my concerns. My concerns have been addressed and therefore, I would like to maintain my original score.

---

> ### Author Response · Authors · 2024-08-13
> **Thank you for your feedback**
>
> Dear Reviewer Dajm,
>
> Thank you for your positive feedback. We will include the new experiments and texts in our revised paper.
>
> Thank you very much for your time and effort!
>
> Best regards,
>
> The Authors

---

### Official Review · Reviewer_fkms · 2024-07-09

**Soundness:** 3
**Presentation:** 2
**Contribution:** 3
**Rating:** 7
**Confidence:** 4

**Summary:**

The authors propose to start with an over-parameterised student model. This is realised using high-order tensors that can reconstruct the original parameter matrices. The idea is that this over-parameterised model will benefit more from knowledge distillation.

**Strengths:**

The ideas is quite interesting/novel for over-parameterising the student during training but preserving the inference parameters (since the higher-order tensors can just be contracted to reconstruct the original weights).

**Weaknesses:**

TinyViT-5M⚗ achieves 80.7% top-1 with 5.4M parameters using a just a distillation token and a logit distillation loss. Comparing this with TinyViT-5M+OPDF 80.0% top-1 and 9.9M/5.4M inference parameters, it is hard to see the benefit? if anything the additional SVD operations and reconstruction losses, make OPDF much more difficult to adopt. What is worse it that this difference is even more significant when going to the larger TinyViT models.

It is understandable that transformers have been the main focus of this work, however it would be good to see some experiments with CNNs. Matrix decomposition for the transformer linear layers is easy (with SVD), but when going to higher-order tensors it will be NP-hard. All the theory presented in this paper is for generalising to arbitrary dimensions, yet all the experiments are done on 2-dimensional weights. The paper would be a lot easier to follow without the pre-mature generalisation. i.e. showing equation (4) in 2-dimensions with SVD and I am not sure if the introduction of the MPO framework is needed.

L177: I am not sure what you mean by "think independantly?". Besides the point that models don't think, if the student can match the teacher fully then that is perfectly fine. The only goal is preserve the performance of the teacher for a much smaller model.

L595: 160 days to pre-train TinyViT is very significant! I understand this is because ImageNet-21K is very large, but why was this chosen benchmark chosen over, for example, a more standard KD benchmark, such as that in DeIT [1] or CRD [2]. Is 160 days an expected scale of time for the number of GPUs used here?

[1] Training data-efficient image transformers & distillation through attention ICML 2021
[2] Contrastive Representation Distillation ICLR 2020

Small points/spelling:
Fig 1a "Batchs -> Batches"

**Questions:**

What is this normalization step in figure 1 and Alg S.1 L9? and its importance? If it is just to preserve the scale and stop exploding values after reconstruction, this could be explained in the text a bit.

**Limitations:**

The authors have adequately addressed these limitations in the checklist.

---

> ### Author Rebuttal · Authors · 2024-08-03
>
> Reply to Reviewer fkms
>
> We sincerely thank you for the constructive comments and suggestions. The following responses will be incorporated into the revised paper.
>
> **Q1. Results on CV tasks are different from original TinyVit paper. Compared to the distillation results of the original method, OPDF does not offer any advantages.**
>
> **Reply:** Great comment! However, we would **clarify** that *the tasks (TinyViT-5M⚗ achieves 80.7\% top-1 with 5.4M parameters) in the original TinyViT paper differ from those in Table 2 of our paper*. In lines 230-231 (please see our paper), we report CV task results based on the performance of models tested directly **without fine-tuning**, whereas the original TinyViT paper presents results **after fine-tuning**.
>
> Typically, the more parameters a pretrained model has, the greater the performance improvement after fine-tuning, which likely explains why larger models exhibit greater performance gaps. In our replication results, our method improved TinyVit's result from 77.4 to 80.0 under the same conditions, demonstrating OPDF's efficacy.
>
> **Q2. Using OPDF in CNN distillation models.**
>
> **Reply:** Thanks for your comment. Ref. [1] affirms MPO's ability to decompose CNNs, confirming its applicability. To show OPDF's effectiveness in CNN distillation, we conducted extra experiments using the methodology from Ref. [2]. This involved distilling a WideResNet from a larger to a smaller version, incorporating OPDF into OFD and classical knowlesge distillation (KD) frameworks. Results and parameters are detailed in **Table 2 and 3** of the *one-page PDF rebuttal file*.
>
> Our results demonstrate that OPDF enhances distillation performance in classical KD and OFD through over-parameterization, underscoring its adaptability in CNN models. This finding will be included in the revised paper. Additionally, as shown in **Table 3** of the *one-page PDF rebuttal file*, OPDF does not increase inference parameters, thus preserving inference time.
>
> **References:**
>
> [1] Z.F. Gao et al. Compressing deep neural networks by matrix product operators. Physical Review Research, 2020, 2(2): 023300.
>
> [2] Heo, B. et al. A comprehensive overhaul of feature distillation. ICCV2019.
>
> **Q3. The difference between MPO and SVD. Necessity and Effectiveness of MPO.**
>
> **Reply:** Thanks for your question. MPO and SVD are different. Since adding dimensions of eigenvalues in SVD does not increase the degrees of freedom, directly expanding the model parameter quantity through SVD improves *very little* of the model capacity. Moreover, the over-parametrization process via SVD is subjected to an upper limit on the number of parameters. Conversely, MPO uses matrices of eigenvectors without expanding eigenvalue dimensions, potentially allowing unlimited model parameter growth.
>
> For example, for a matrix ${W} \in \mathcal{R}^{I\times J}$, the parameter limit via SVD is $I^2 + J^2$. In contrast, over-parameterizing ${W}$ using MPO with $n$ tensors allows more parameters, as given by follwing equation, proving more effective than SVD.
>
> $$N = \sum_{k=1}^{m} i_kj_kd_{k-1}d_k.$$
>
> Besides allowing the decomposition of the parameter matrix into an arbitrary number of effective parameters, OPDF demonstrates more significant performance improvements compared to SVD, as detailed in Section 5.2 of our pape. According to Tables 1 and 2 in our paper, while incorporating SVD generally enhances the knowlesge distillation (KD) model's performance across most datasets, it does not perform as well as OPDF.
>
> **Q4. The meaning of "think independantly".**
>
> **Reply:** Great remark! We will rephrase this statement to be "learn independently". In the OPDF, auxiliary tensor alignment allows the student model to learn task-relevant information. Through distillation with the central tensor, the student model not only can imitate the teacher model, but also equips itself with the capability to learn independently from the original labels, detached from the teacher model. Consequently, this endows the model with the potential to surpass the teacher model.
>
> Conversely, if the parameter matrix is fully aligned, the student model merely emulates the teacher model. Experimental results validate this, with Table 1 showing that after adopting OPDF, the student model exceeds the teacher by 1.2% on the RTE dataset.
>
> **Q5. The training time for TinyViT is excessively long.**
>
> **Reply:** Excellent comment! One GPU day refers to *running a single GPU for one day*, and actual distillation time can be calculated using multi-GPU parallelism. By utilizing four parallel servers (each equipped with 8 NVIDIA A100 GPUs) for training, the actual training time needed amounts to 5 days. Initially, the TinyViT paper stated a distillation time of 140 GPU days, which increased to 160 GPU days (14% rise) after using the OPDF method, significantly enhancing effectiveness.
>
> Table 1 of our paper shows the parameter changes after parameterization, e.g., DBKD(53M) *vs.* DBKD+OPDF(83M), with performance increased from 74.4 to 77.6. It is evident that introducing additional training parameters incurs a certain training cost increase, yet it notably boosts the distillation performance.
>
> **Q6. The meaning and significance of normalization.**
>
> **Reply:** Great question! During MPO decomposition, normalization distributes information evenly across tensors. As depicted in Algorithm 1 in our paper, after $n$ SVD iterations, all eigenvalue information accumulates in the final tensor, while the preceding $n-1$ tensors are derived from unitary matrix decompositions and do not contain significant information. Normalization effectively distribute information across each tensor.
>
> **Concluding remark:** We sincerely thank you for putting forward excellent comments. We hope the above responses are helpful to clarify your questions. We look forward to addressing any additional questions. Your consideration of improving the rating of our paper will be much appreciated!

---

> ### Author Response · Authors · 2024-08-10
> **Looking forward to your feedback**
>
> Dear Reviewer fkms,
>
> We're keen to know if there are any remaining concerns that require attention or if there are additional discussions that should take place. Your insights are greatly appreciated as they contribute to the refinement of the paper. Looking forward to your feedback. Thank you.
>
> Best regards,
>
> The Authors

---

> ### Comment · Reviewer_fkms · 2024-08-11
>
> Thank you for your thorough response and additional experiments. Most of my concerns have been addressed and in light of the other reviewers remarks I am happy to raise my score. However, I am still unsure about 2 main parts.
>
> Firstly, although I do understand and appreciate that the authors only report the linear probe performance and thus a comparison to TinyViT, DeiT etc is unfair, I would like to know why this is done? The authors don't seem to do this for the NLP related tasks. Are there other KD works that train models and evaluate in this fashion? (specifically for vision-tasks). Vision KD is a lot more more mature than NLP KD, especially for feature distillation, and so I believe a comparison here is important.
>
> Secondly, the experiments on CNN models is greatly appreciated and does strengthen this submission. It is important to show that the generalisation has practical utility rather than just for the sake of generality. The authors have addressed my concern over this point theoretically too. These CIFAR100 results are fully fine-tuned which is nice to see and partly addresses my previous concern. OFD is argueably quite an old paper. Is there any chance for a comparison to some other architecture pairs provided in the CRD [1] benchmark? and using some more recent KD methods. There are some simple methods with code here [2].
>
> Hope the authors can address my concerns.
>
> [1] "Contrastive Representation Distillation" ICLR 2020
> [2] https://paperswithcode.com/sota/knowledge-distillation-on-cifar-100

---

> ### Author Response · Authors · 2024-08-12
> **Additional vision KD results (part 1)**
>
> We greatly appreciate your feedback along with two additional questions. Please see our reply to your *first question* as follows.
>
> **Q1. The application of linear probe performance metrics; distillation results affter fine-tuining.**
>
> **Reply:** Thanks for your question. In fact, Table 5 in original TinyVit paper [1] has already utilized linear probe performance to assess the efficacy of TinyVit. Hence, we opted it for comparison to effectively demonstrate the validity of OPDF.
>
> Moreover, we have fine-tuned models previously distilled following the tasks set in TinyViT (where TinyViT-5M⚗ achieves 80.7% top-1 accuracy with 5.4M parameters), and the results are showed in **Table A** below to show the efficiency of our proposed OPDF.
>
> The experiments results show that even when employing the same experimental setup in TinyVit, OPDF consistently yields a significant enhancement in the performance of TinyVit. We will include these results in the revised paper.
>
> **Table A.** Models are pretrained on ImageNet-21k and then finetuned on ImageNet-1k, Imagenet-Real and Imagenet-V2 (top-1/top-5).
> | Datasets  | Imagenet-1k | Imagenet-Real | Imagenet-V2 |  # Train Params | # Inference Params |
> | --------- | :--------: | :--------:| :--------:| :--------: | :-----------: |
> | TinyVit-5M      | 80.7/95.6 | 87.5/97.8 | 68.3/89.7 |  5.4 | 5.4    |
> | TinyVit-5M+OPDF  |**81.8/96.9** | **87.9/98.4** | **69.5/90.4** | 9.9  |   5.4  |
> | TinyVit-11M     | 83.2/96.5 | 88.3/98.1 | 72.9/91.4 |  11  |   11   |
> | TinyVit-11M+OPDF| **85.1/97.1** | **89.0/98.5** | **74.1/93.3** | 23   |  11    |
> | TinyVit-21M     | 84.8/97.3 | 88.9/98.5 | 75.1/93.5 |  21  |    21  |
> | TinyVit-21M+OPDF| **86.5/97.9** | **89.7/98.9** | **76.2/94.7** | 38   |   21   |
>
> BTW, the code of applying OPDF to other recent CNN KD methods (e.g., CRD) is currently executing. We will supplement the corresponding results as soon as the model training is completed. Hence, we will respond to your *second question* soon. Thank you for your patience.
>
> **References:**
>
> [1] Wu, et al. TinyViT: Fast Pretraining Distillation for Small Vision Transformers. ECCV 2022, pages 68–85.

---

> > ### Comment · Reviewer_fkms · 2024-08-13
> >
> > Thank you for your extensive effort in response to my questions. I am happy to raise my score to 5 BA, and I am also very keen to see what will come back with regards to using other recent CNN KD methods.

---

> > > ### Author Response · Authors · 2024-08-13
> > > **Additional vision KD results (part 2)**
> > >
> > > The following response reports the additional KD results for more recent vision models.
> > >
> > > **Q2.The Performance of OPDF on some architecture pairs provided in CRD and several recent CNN KD methods.**
> > >
> > > **Reply:** Thanks for your great question. We have applied OPDF to several architecture pairs in CRD and recent KD methods to show the capability of our method in enhancing the performance of CNN distillation models. The new results are reported in **Table B**.
> > >
> > > It can be seen that our proposed OPDF can enhance student model performance in several architecture pairs provided in the CRD benchmark using more recent KD methods. We will include these results in the revised paper.
> > >
> > > **Table B.** Distillation results of various CNN architectures on CIFAR-100.
> > >
> > > | Teacher to Student  | ResNet110 to ResNet20 | ResNet110 to ResNet32 | ResNet32x4 to ResNet8x4 |
> > > | --------- | :--------: | :--------:| :--------:|
> > > | Teacher    | 74.31 | 74.31 | 79.42 |
> > > | CRD        | 71.46 | 73.48 | 75.51 |
> > > | CRD+OPDF   | **72.35** | **74.57** | **75.83** |
> > > | NST        | 69.53 | 71.96 | 73.30 |
> > > | NST+OPDF   | **70.98** | **72.34** | **74.00** |
> > > | ITRD [1] | 71.99 | 74.26 | 76.19 |
> > > | ITRD+OPDF  | **72.48** | **75.92** | **77.73** |
> > > | KD+LSKD [2]    | 71.99 | 74.26 | 76.19 |
> > > | KD+LSKD+OPDF  | **72.33** | **74.93** |**76.89** |
> > >
> > >
> > > Since the discussion period is soon ending, this is the best result we could supply at this moment after nightless working on the new experiments.
> > >
> > > Thank you very much for your time and efforts. Your consideration of further improving the rating of our paper will be much appreciated!
> > >
> > >
> > >
> > > **Reference:**
> > >
> > > [1] Miles, et al. Information theoretic representation distillation. In British Machine Vision Conference (BMVC), 2022.
> > >
> > > [2] Sun, et al. Logit standardization in knowledge distillation. In Proceedings of the IEEE/CVF Conference on Computer Vision and Pattern Recognition, pages 15731–15740, 2024.

---

> > > > ### Comment · Reviewer_fkms · 2024-08-13
> > > >
> > > > These results look really great and it would be good to see them in the updated manuscript. I will happily raise my score. The authors have done a great job at addressing all my concerns and it is very clear that their proposed method is complimentary and easy to integrate into many KD pipelines.

---

> > > > > ### Author Response · Authors · 2024-08-13
> > > > > **Thank you for your positive feedback**
> > > > >
> > > > > Dear Reviewer fkms,
> > > > >
> > > > > We greatly appreciate your positive feedback. The discussion with you has been quite productive and fruitful. Thank you very much for raising the score. We will include all the new experiment results and discussions in the revised paper.
> > > > >
> > > > > Thank you for your time and effort!
> > > > >
> > > > > Best regards,
> > > > >
> > > > > The Authors

---

### Official Review · Reviewer_Qtmi · 2024-07-12

**Soundness:** 3
**Presentation:** 3
**Contribution:** 3
**Rating:** 6
**Confidence:** 4

**Summary:**

This paper proposes a novel over-parameterization framework designed to enhance the effectiveness of knowledge distillation. This framework employs MPO as a tensor decomposition technique to expand small models into larger ones to give the student model more capacity. Moreover, to enhance the effectiveness of knowledge distillation, a tensor constraint loss is introduced to align the teacher and student model. Extensive experiments verify the superiority of the method.

**Strengths:**

1. Enhancing the capacity of the student model through tensor decomposition is novel and it does not incur additional inference overhead.
2. The loss constraint for aligning the auxiliary tensors between the student and teacher models is also quite different from the conventional logit or feature matching, representing a new approach of matching in KD.
3. The experiments are comprehensive; the authors test the method on many benchmarks in both CV and NLP, proving the effectiveness of the method.

**Weaknesses:**

1. I would like to know how much additional distillation cost (memory and time cost) will be incurred by introducing such tensor decomposition technique and loss constraint for aligning the auxiliary tensors between the student and teacher models.
2. Especially in the era of LLM, the high cost of distillation often limits its application. Methods like LoRA have been proposed to reduce the trainable model parameters. I am concerned that if this approach goes against the current mainstream research directions?
3. Can the authors provide some distillation results on larger models to further validate the applicability of the approach on large models?

**Questions:**

Please refer to weaknesses.

---

> ### Author Rebuttal · Authors · 2024-08-03
>
> We sincerely thank you for the constructive comments and suggestions, which are very helpful for improving our paper. We are also grateful that you recognized the strengths and contributions of our paper. Moreover, the following responses will be incorporated into the revised paper.
>
>
> **Q1. Additional distillation cost (memory and time) incurred by introducing OPDF.**
>
> **Reply:** Great remark! We listed the distillation cost (memory and time cost) of the original model and the model after applying OPDF in **Table 1** in the *one-page PDF rebuttal file*. We can observe that as the number of parameters obtained from MPO decomposition increases, both the training time and memory cost increase. However, as the dataset size increases, the ratio of additional time and memory required for training by OPDF  to the original training requirements generally exhibits a decreasing trend (e.g., 0.6/0.4 for RTE *vs* 0.3/0.1 for MNLI in BERT-of-Theseus model). Therefore, the additional time and memory introduced by our method become less of a critical bottleneck affecting the training speed as the dataset size increases. Hope this clarifies your question.
>
>
> **Q2. OPDF may conflict with mainstream research directions aimed at reducing trainable model parameters (e.g., LoRA).**
>
> **Reply:** Excellent comment! We would like to clarify that our approach has different goal compared with LoRA. Whereas LoRA is designed to reduce the number of parameters during the lightweight fine-tuning process, OPDF aims to enhance the capabilities of existing knowledge distillation models through an over-parameterization procedure.
>
> Nonetheless, we performed tests to assess the effect of over-parameterization through MPO on model performance during the lightweight fine-tuning process. The results are presented in **Table A**. We observe that the implementation of MPO can augment the efficacy of BERT-base models without extending the inference duration or expanding the parameter volume. Conversely, the adoption of LoRA necessitates the parameter volume for inference, which consequently lengthens the inference time.
>
> **Table A.** The result of lightweight fine-tuning on GLUE by using MPO and LoRA.
>
> | Datasets  | RTE | MRPC | STS-B | CoLA | SST-2 | QNLI | QQP | MNLI | Avg.     | # Train Params | # Inference Params |
> | --------- | :--------: | :--------: | :-----------: | :---------: | :----------: | :---------: | :--------: | :---------: | :--------: | :--------------: | :------------------: |
> | BERT-base | 70.5     | 86.5     | 86.6        | 54.2      | 92.0       | 91.2      | 91.0     | 84.2      | 82.0     | 110M           | 110M               |
> | LoRA      | 71.5     | 89.8     | 88.6        | 58.3      | 91.5       | 90.3      | 91.7     | 83.3      | 83.1     | 295K           | 110M+295K          |
> | +MPO      | **72.3** | **90.0** | **89.0**    | **60.6**  | **92.5**   | **91.5**  | **92.3** | **85.1**  | **83.7** | 341M           | 110M               |
>
>
> **Q3. Distillation results on larger models.**
>
> **Reply:** Great comment! We have implemented OPDF on the GPT-2-760M, OPT-6.7B and LLAMA-7B models, with the corresponding teacher models of GPT-2-1.5B, OPT-13B and LLAMA-13B, respectively. We have reported the Rouge-L scores of these models on 5 instruction-following datasets, with the results displayed in **Table B**. We can observe that, after implementing OPDF, the efficiency of distillation on large models is improved across all datasets. This demonstrates that our method is also highly effective on larger models.
>
> **Table B.** Distillation results on larger models with OPDF.
>
> | Model     | #Params | Method  | Dolly    | SelfInst | Vicuna   | S-NI     | UnNI     | Avg.     | # Train Params | # Inference Params |
> | --------- | :-------: | :-------: | :--------: | :--------: | :--------: | :--------: | :--------: | :--------: | :--------------: | :------------------: |
> | **GPT-2** | 1.5B    | Teacher | 27.6     | 14.3     | 16.3     | 27.6     | 31.8     | 23.5     | 1.5B           | 1.5B               |
> | **GPT-2** | 760M    | w/o KD  | 25.4     | 12.4     | 16.1     | 21.5     | 24.0     | 19.9     | 760M           | 760M               |
> | **GPT-2** | 760M    | KD      | 25.9     | 13.4     | 16.9     | 25.3     | 28.0     | 21.9     | 760M           | 760M               |
> | **GPT-2** | 760M    | KD+OPDF | **26.1** | **14.1** | **17.5** | **25.7** | **28.6** | **22.4** | 1.3B           | 760M               |
> | **OPT**   | 13B     | Teacher | 29.2     | 18.4     | 17.8     | 30.4     | 36.1     | 26.4     | 13B            | 13B                |
> | **OPT**   | 6.7B    | w/o KD  | 27.6     | 16.4     | 17.8     | 30.3     | 28.6     | 24.1     | 6.7B           | 6.7B               |
> | **OPT**   | 6.7B    | KD      | 28.3     | 17.0     | 17.5     | 30.7     | 26.7     | 24.0     | 6.7B           | 6.7B               |
> | **OPT**   | 6.7B    | KD+OPDF | **28.5** | **17.7** | **17.9** | **31.4** | **29.8** | **25.1** | 14B            | 6.7B               |
> | **LLaMA** | 13B     | Teacher | 29.7     | 23.4     | 19.4     | 35.8     | 38.5     | 29.4     | 13B            | 13B                |
> | **LLaMA** | 7B      | w/o KD  | 26.3     | 20.8     | 17.5     | 32.4     | 35.8     | 26.6     | 7B             | 7B                 |
> | **LLaMA** | 7B      | KD      | 27.4     | 20.2     | 18.4     | 33.7     | 37.9     | 27.5     | 7B             | 7B                 |
> | **LLaMA** | 7B      | KD+OPDF | **27.5** | **21.6** | **19.7** | **34.8** | **40.0** | **28.7** | 10B            | 7B                 |
>
>
> **Concluding remark:** We sincerely thank you for reviewing our paper and putting forward thoughtful comments/suggestions. We hope the above responses are helpful to clarify your questions. We will be happy to hear your feedback and look forward to addressing any additional questions. Your consideration of improving the rating of our paper will be much appreciated!

---

> > ### Comment · Reviewer_Qtmi · 2024-08-12
> >
> > I am confused about why the authors claimed that the inference parameters of adding LoRA will increase. What I mean is that performing KD on LLMs is expensive. Reducing the number of trainable parameters might be a way to lower the cost of performing KD.
> >
> > However, since the authors have presented the time and memory cost of the proposed method and it seems to be acceptable, I would like to raise my score by 1.

---

> ### Author Response · Authors · 2024-08-10
> **Looking forward to your feedback**
>
> Dear Reviewer Qtmi,
>
> We're keen to know if there are any remaining concerns that require attention or if there are additional discussions that should take place. Your insights are greatly appreciated as they contribute to the refinement of the paper. Looking forward to your feedback. Thank you.
>
> Best regards,
>
> The Authors

---

> ### Author Response · Authors · 2024-08-12
> **Thank you for raising the score**
>
> Thank you for your positive response. We would like to clarify that during the inference with LoRA, the parameters that need to be computed are $W+\Delta W$, where $W$ represents the weights of the pre-trained model (PLM) and $\Delta W$ the additional parameters introduced by LoRA. This is why we mentioned in our reply that applying the LoRA method during inference increases the number of parameters, thereby increasing the inference time.
>
> Lightweight fine-tuning (LoRA) and model compression (KD) represent two different tracks for deploying large PLM. While LoRA is cost-effective, it slightly increases the inference time due to the added $\Delta W$ parameters. On the other hand, KD results in a compressed model that reduces both inference time and computational overhead while increasing the training cost.
>
> Moreover, compressed models obtained through KD may have diverse applications on the edge. For instance, they are particularly suitable for deployment in resource-constrained environments, such as mobile devices and embedded systems.
>
> Again, thank you very much for increasing the rating of our paper!

---

### Author Rebuttal · Authors · 2024-08-06

Global Response:

Dear Reviewers:

We would like to thank you for your constructive comments, which are very helpful in improving our paper. We have posted the point-to-point reply to each question/comment raised by you. And we have listed three additional tables in the *one-page PDF rebuttal file*, which contained "Training time and Memory Cost for OPDF", "Distillation results on various CNN architectures with OPDF" and " Train Params and Inference Params of CNN model". Please do feel free to let us know if you have any further questions.

We are also pleased that the reviewers recognized the novelty and versatility of our work. In particular, we thank the reviewers for recognizing the *preservation of the inference parameters* (Qtmi and fkms), *adequacy of the experiment* (Dajm), and *novelty* (XV5e) of our method.

Thank you very much.

Best regards,

The Authors of the Paper

---

### Decision · Program_Chairs · 2024-09-25

**Decision:**

Accept (poster)

**Comment:**

The paper presents a novel framework for enhancing knowledge distillation (KD) by leveraging over-parameterization. Using the Matrix Product Operator (MPO) as a tensor decomposition technique, the authors propose to increase the capacity of the student model during training, improving its ability to learn from the teacher model without adding inference overhead. Additionally, a loss is introduced to align the student and teacher models more effectively. Extensive experiments across various vision and NLP benchmarks showcase the utility of the proposed KD framework.

Reviewers appreciated the innovative approach, particularly the use of tensor decomposition to over-parameterize the student model. The introduction of a tensor constraint loss for better alignment between teacher and student models was also highlighted as a significant strength. The comprehensive experimental validation was seen as a key strength. However, some concerns were raised about the clarity of the paper, particularly in explaining certain concepts, and the scalability and computational cost of the proposed method, especially for larger models. Despite these concerns, I will recommended the paper for acceptance and suggest the authors incorporate the feedback.